# Sum-of-Squares Lower Bounds for Sparse PCA

Tengyu Ma[*1] and Avi Wigderson[†2]

[1]Department of Computer Science, Princeton University
[2]School of Mathematics, Institute for Advanced Study

## Abstract

This paper establishes a statistical versus computational trade-off for solving a basic high-dimensional machine learning problem via a basic convex relaxation method. Specifically, we consider the *Sparse Principal Component Analysis* (Sparse PCA) problem, and the family of *Sum-of-Squares* (SoS, aka Lasserre/Parillo) convex relaxations. It was well known that in large dimension $p$, a planted $k$-sparse unit vector can be *in principle* detected using only $n \approx k \log p$ (Gaussian or Bernoulli) samples, but all *efficient* (polynomial time) algorithms known require $n \approx k^2$ samples. It was also known that this quadratic gap cannot be improved by the the most basic *semi-definite* (SDP, aka spectral) relaxation, equivalent to a degree-2 SoS algorithms. Here we prove that also degree-4 SoS algorithms cannot improve this quadratic gap. This average-case lower bound adds to the small collection of hardness results in machine learning for this powerful family of convex relaxation algorithms. Moreover, our design of moments (or "pseudo-expectations") for this lower bound is quite different than previous lower bounds. Establishing lower bounds for higher degree SoS algorithms for remains a challenging problem.

## 1 Introduction

We start with a general discussion of the tension between sample size and computational efficiency in statistical and learning problems. We then describe the concrete model and problem at hand: Sum-of-Squares algorithms and the Sparse-PCA problem. All are broad topics studied from different viewpoints, and the given references provide more information.

### 1.1 Statistical vs. computational sample-size

Modern machine learning and statistical inference problems are often high dimensional, and it is highly desirable to solve them using far less samples than the ambient dimension. Luckily, we often know, or assume, some underlying structure of the objects sought, which allows such savings *in principle*. Typical such assumption is that the number of *real* degrees of freedom is far smaller than the dimension; examples include sparsity constraints for vectors, and low rank for matrices and tensors. The main difficulty that occurs in nearly all these problems is that while information theoretically the sought answer is present (with high probability) in a small number of samples, actually computing (or even approximating) it from these many samples is a computationally hard problem. It is often expressed as a non-convex optimization program which is NP-hard in the worst case, and seemingly hard even on random instances.

Given this state of affairs, *relaxed* formulations of such non-convex programs were proposed, which can be solved efficiently, but sometimes to achieve accurate results seem to require far more samples

---

[*]Supported in part by Simons Award for Graduate Students in Theoretical Computer Science
[†]Supported in part by NSF grant CCF-1412958

than existential bounds provide. This phenomenon has been coined the "statistical versus computational trade-off" by Chandrasekaran and Jordan [1], who motivate and formalize one framework to study it in which efficient algorithms come from the Sum-of-Squares family of convex relaxations (which we shall presently discuss). They further give a detailed study of this trade-off for the basic *de-noising problem* [2, 3, 4] in various settings (some exhibiting the trade-off and others that do not). This trade-off was observed in other practical machine learning problems, in particular for the Sparse PCA problem that will be our focus, by Berthet and Rigollet [5].

As it turns out, the study of the same phenomenon was proposed even earlier in computational complexity, primarily from theoretical motivations. Decatur, Goldreich and Ron [6] initiate the study of "computational sample complexity" to study statistical versus computation trade-offs in sample-size. In their framework efficient algorithms are arbitrary polynomial time ones, not restricted to any particular structure like convex relaxations. They point out for example that in the distribution-free PAC-learning framework of Vapnik-Chervonenkis and Valiant, there is often no such trade-off. The reason is that the number of samples is essentially determined (up to logarithmic factors, which we will mostly ignore here) by the VC-dimension of the given concept class learned, and moreover, an "Occam algorithm" (computing *any* consistent hypothesis) suffices for classification from these many samples. So, in the many cases where efficiently finding a hypothesis consistent with the data is possible, enough samples to learn are enough to do so efficiently! This paper also provide examples where this is not the case in PAC learning, and then turns to an extensive study of possible trade-offs for learning various concept classes under the uniform distribution. This direction was further developed by Servedio [7].

The fast growth of Big Data research, the variety of problems successfully attacked by various heuristics and the attempts to find efficient algorithms with provable guarantees is a growing area of interaction between statisticians and machine learning researchers on the one hand, and optimization and computer scientists on the other. The trade-offs between sample size and computational complexity, which seems to be present for many such problems, reflects a curious "conflict" between these fields, as in the first more data is good news, as it allows more accurate inference and prediction, whereas in the second it is bad news, as a larger input size is a source of increased complexity and inefficiency. More importantly, understanding this phenomenon can serve as a guide to the design of better algorithms from both a statistical and computational viewpoints, especially for problems in which data acquisition itself is costly, and not just computation. A basic question is thus for which problems is such trade-off inherent, and to establish the limits of what is achievable by efficient methods.

Establishing a trade-off has two parts. One has to prove an existential, information theoretic upper bound on the number of samples needed when efficiency is not an issue, and then prove a computational lower bound on the number of samples for the class of efficient algorithms at hand. Needless to say, it is desirable that the lower bounds hold for as wide a class of algorithms as possible, and that it will match the best known upper bound achieved by algorithms from this class. The most general one, the computational complexity framework of [6, 7] allows all polynomial-time algorithms. Here one cannot hope for unconditional lower bounds, and so existing lower bounds rely on computational assumptions, e.g."cryptographic assumptions", e.g. that factoring integers has no polynomial time algorithm, or other average case assumptions. For example, hardness of refuting random 3CNF was used for establishing the sample-computational tradeoff for learning halfspaces [8], and hardness of finding planted clique in random graphs was used for tradeoff in sparse PCA [5, 9]. On the other hand, in frameworks such as [1], where the class of efficient algorithms is more restricted (e.g. a family of convex relaxations), one can hope to prove unconditional lower bounds, which are called "integrality gaps" in the optimization and algorithms literature. Our main result is of this nature, adding to the small number of such lower bounds for machine learning problems.

We now describe and motivate SoS convex relaxations algorithms, and the Sparse PCA problem.

## 1.2 Sum-of-Squares convex relaxations

Sum-of-Squares algorithms (sometimes called the Lasserre hierarchy) encompasses perhaps the strongest known algorithmic technique for a diverse set of optimization problems. It is a family of convex relaxations introduced independently around the year 2000 by Lasserre [10], Parillo [11], and in the (equivalent) context of proof systems by Grigoriev [12]. These papers followed better and better understanding in real algebraic geometry [13, 14, 15, 16, 17, 18, 19]of David Hilbert's

famous 17th problem on certifying the non-negativity of a polynomial by writing it as a *sum of squares* (which explains the name of this method). We only briefly describe this important class of algorithms; far more can be found in the book [20] and the excellent extensive survey [21].

The SoS method provides a principled way of adding constraints to a linear or convex program in a way that obtains tighter and tighter convex sets containing all solutions of the original problem. This family of algorithms is parametrized by their *degree* $d$ (sometimes called the *number of rounds*); as $d$ gets larger, the approximation becomes better, but the running time becomes slower, specifically $n^{O(d)}$. Thus in practice one hopes that small degree (ideally constant) would provide sufficiently good approximation, so that the algorithm would run in polynomial time. This method extends the standard semi-definite relaxation (SDP, sometimes called spectral), that is captured already by degree-2 SoS algorithms. Moreover, it is more powerful than two earlier families of relaxations: the Sherali-Adams [22] and Lovász-Scrijver [23] hierarchies.

The introduction of these algorithms has made a huge splash in the optimization community, and numerous applications of it to problems in diverse fields were found that greatly improve solution quality and time performance over all past methods. For large classes of problems they are considered the strongest algorithmic technique known. Relevant to us is the very recent growing set of applications of constant-degree SoS algorithms to machine learning problems, such as [24, 25, 26]. The survey [27] contains some of these exciting developments. Section 2.1 contains some self-contained material about the general framework SoS algorithms as well.

Given their power, it was natural to consider proving lower bounds on what SoS algorithms can do. There has been an impressive progress on SoS degree lower bounds (via beautiful techniques) for a variety of combinatorial optimization problems [28, 12, 29, 30]. However, for machine learning problems relatively few such lower bounds (above SDP level) are known [26, 31] and follow via reductions to the above bounds. So it is interesting to enrich the set of techniques for proving such limits on the power of SoS for ML. The lower bound we prove indeed seem to follow a different route than previous such proofs.

### 1.3   Sparse PCA

Sparse principal component analysis, the version of the classical PCA problem which assumes that the direction of variance of the data has a sparse structure, is by now a central problem of high-dimensional statistical analysis. In this paper we focus on the single-spiked covariance model introduced by Johnstone [32]. One observes $n$ samples from $p$-dimensional Gaussian distribution with covariance $\Sigma = \lambda vv^T + I$ where (the *planted* vector) $v$ is assumed to be a unit-norm *sparse* vector with at most $k$ non-zero entries, and $\lambda > 0$ represents the strength of the signal. The task is to find (or estimate) the sparse vector $v$. More general versions of the problem allow several sparse directions/components and general covariance matrix [33, 34]. Sparse PCA and its variants have a wide variety of applications ranging from signal processing to biology: see, e.g., [35, 36, 37, 38].

The hardness of Sparse PCA, at least in the worst case, can be seen through its connection to the (NP-hard) Clique problem in graphs. Note that if $\Sigma$ is a $\{0, 1\}$ adjacency matrix of a graph (with 1's on the diagonal), then it has a $k$-sparse eigenvector $v$ with eigenvalue $k$ if and only if the graph has a $k$-clique. This connection between these two problems is actually deeper, and will appear again below, for our real, average case version above.

From a theoretical point of view, Sparse PCA is one of the simplest examples where we observe a gap between the number of samples needed information theoretically and the number of samples needed for a polynomial time estimator: It has been well understood [39, 40, 41] that information theoretically, given $n = O(k \log p)$ samples[1], one can estimate $v$ up to constant error (in euclidean norm), using a non-convex (therefore not polynomial time) optimization algorithm. On the other hand, all the existing provable polynomial time algorithms [36, 42, 34, 43], which use either diagonal thresholding (for the single spiked model) or semidefinite programming (for general covariance), first introduced for this problem in [44], need at least quadratically many samples to solve the problem, namely $n = O(k^2)$. Moreover, Krauthgamer, Nadler and Vilenchik [45] and Berthet and Rigollet [41] have shown that for semi-definite programs (SDP) this bound is tight. Specifically, the natural SDP cannot even solve the *detection problem*: to distinguish the data from covariance

$\Sigma = \lambda vv^T + I$ from the null hypothesis in which no sparse vector is planted, namely the $n$ samples are drawn from the Gaussian distribution with covariance matrix $I$.

Recall that the natural SDP for this problem (and many others) is just the first level of the SoS hierarchy, namely degree-2. Given the importance of the Sparse PCA, it is an intriguing question whether one can solve it efficiently with far fewer samples by allowing degree-$d$ SoS algorithms with larger $d$. A very interesting *conditional* negative answer was suggested by Berthet and Rigollet [41]. They gave an efficient reduction from *Planted Clique*[2] problem to Sparse PCA, which shows in particular that degree-$d$ SoS algorithms for Sparse PCA will imply similar ones for Planted Clique. Gao, Ma and Zhou [9] strengthen the result by establishing the hardness of the Gaussian single-spiked covariance model, which is an interesting subset of models considered by [5]. These are useful as nontrivial constant-degree SoS lower bounds for Planted Clique were recently proved by [30, 46] (see there for the precise description, history and motivation for Planted Clique). As [41, 9] argue, strong yet *believed* bounds, if true, would imply that the quadratic gap is tight for any constant $d$. Before the submission of this paper, the known lower bounds above for planted clique were not strong enough yet to yield any lower bound for Sparse PCA beyond the minimax sample complexity. We also note that the recent progress [47, 48] that show the tight lower bounds for planted clique, together with the reductions of [5, 9], also imply the tight lower bounds for Sparse PCA, as shown in this paper.

## 1.4  Our contribution

We give a direct, unconditional lower bound proof for computing Sparse PCA using degree-4 SoS algorithms, showing that they too require $n = \widetilde{\Omega}(k^2)$ samples to solve the detection problem (Theorem 3.1), which is tight up to polylogarithmic factors when the strength of the signal $\lambda$ is a constant. Indeed the theorem gives a lower bound for every strength $\lambda$, which becomes weaker as $\lambda$ gets larger. Our proof proceeds by constructing the necessary pseudo-moments for the SoS program that achieve too high an objective value (in the jargon of optimization, we prove an "integrality gap" for these programs). As usual in such proofs, there is tension between having the pseudo-moments satisfy the constraints of the program and keeping them positive semidefinite (PSD). Differing from past lower bound proofs, we construct *two* different PSD moments, each approximately satisfying one sets of constraints in the program and is negligible on the rest. Thus, their sum give PSD moments which *approximately* satisfy all constraints. We then perturb these moments to satisfy constraints *exactly*, and show that with high probability over the random data, this perturbation leaves the moments PSD.

We note several features of our lower bound proof which makes the result particularly strong and general. First, it applies not only for the Gaussian distribution, but also for Bernoulli and other distributions. Indeed, we give a set of natural (pseudorandomness) conditions on the sampled data vectors under which the SoS algorithm is "fooled", and show that these conditions are satisfied with high probability under many similar distributions (possessing strong concentration of measure). Next, our lower bound holds even if the hidden sparse vector is discrete, namely its entries come from the set $\{0, \pm\frac{1}{\sqrt{k}}\}$. We also extend the lower bound for the detection problem to apply also to the estimation problem, in the regime when the ambient dimension is linear in the number of samples, namely $n \leq p \leq Bn$ for constant $B$.

**Organization:** Section 2 provides more backgrounds of sparse PCA and SoS algorithms. We state our main results in Section 3. A complete paper is available as supplementary material or on arxiv.

## 2  Formal description of the model and problem

**Notation:** We will assume that $n, k, p$ are all sufficiently large[3], and that $n \leq p$. Throughout this paper, by "with high probability some event happens", we mean the failure probability is bounded by $p^{-c}$ for every constant $c$, as $p$ tends to infinity.

**Sparse PCA estimation and detection problems**   We will consider the simplest setting of sparse PCA, which is called single-spiked covariance model in literature [32] (note that restricting to a

special case makes our lower bound hold in all generalizations of this simple model). In this model, the task is to recover a single sparse vector from noisy samples as follows. The "hidden data" is an unknown $k$-sparse vector $v \in \mathbb{R}^p$ with $|v|_0 = k$ and $\|v\| = 1$. To make the task easier (and so the lower bound stronger), we even assume that $v$ has discrete entries, namely that $v_i \in \{0, \pm\frac{1}{\sqrt{k}}\}$ for all $i \in [p]$. We observe $n$ noisy samples $X^1, \ldots, X^n \in \mathbb{R}^p$ that are generated as follows. Each is independently drawn as $X^j = \sqrt{\lambda} g^j v + \xi^j$ from a distribution which generalizes both Gaussian and Bernoulli noise to $v$. Namely, the $g^j$'s are i.i.d real random variable with mean 0 and variance 1, and $\xi^j$'s are i.i.d random vectors which have independent entries with mean zero and variance 1. Therefore under this model, the covariance of $X^i$ is equal to $\lambda v v^T + I$. Moreover, we assume that $g^j$ and entries of $\xi^j$ are sub-gaussian[4] with variance proxy $O(1)$. Given these samples, the *estimation* problem is to approximate the unknown sparse vector $v$ (up to sign flip).

It is also interesting to also consider the sparse component *detection* problem [41, 5], which is the decision problem of distinguishing from random samples the following two distributions

$H_0$: data $X^j = \xi^j$ is purely random

$H_v$: data $X^j = \xi^j + \sqrt{\lambda} g^j v$ contains a hidden sparse signal with strength $\lambda$.

Rigollet [49] observed that a polynomial time algorithm for estimation version of sparse PCA with constant error implies that an algorithm for the detection problem with twice number of the samples. Thus, for polynomial time lower bounds, it suffices to consider the detection problem. We will use $X$ as a shorthand for the $p \times n$ matrix $[X^1, \ldots, X^n]$. We denote the rows of $X$ as $X_1^T, \ldots, X_p^T$, therefore $X_i$'s are $n$-dimensional column vectors. The empirical covariance matrix is defined as $\hat{\Sigma} = \frac{1}{n} X X^T$.

**Statistically optimal estimator/detector**     It is well known that the following non-convex program achieves optimal statistical minimax rate for the estimation problem and the optimal sample complexity for the detection problem. Note that we scale the variables $x$ up by a factor of $\sqrt{k}$ for simplicity (the hidden vector now has entries from $\{0, \pm 1\}$).

$$\lambda_{\max}^k(\hat{\Sigma}) = \frac{1}{k} \cdot \max \qquad \langle \hat{\Sigma}, x x^T \rangle \tag{2.1}$$

$$\text{subject to} \qquad \|x\|_2^2 = k, \|x\|_0 = k \tag{2.2}$$

**Proposition 2.1** ([42], [41], [39] informally stated). The non-convex program (2.1) statistically optimally solves the sparse PCA problem when $n \geq Ck/\lambda^2 \log p$ for some sufficiently large $C$. Namely, the following hold with high probability. If $X$ is generated from $H_v$, then optimal solution $x_{\text{opt}}$ of program (2.1) satisfies $\|\frac{1}{k} \cdot x_{\text{opt}} x_{\text{opt}}^T - v v^T\| \leq \frac{1}{3}$, and the objective value $\lambda_{\max}^k(\hat{\Sigma})$ is at least $1 + \frac{2\lambda}{3}$. On the other hand, if $X$ is generated from null hypothesis $H_0$, then $\lambda_{\max}^k(\hat{\Sigma})$ is at most $1 + \frac{\lambda}{3}$.

Therefore, for the detection problem, once can simply use the test $\lambda_{\max}^k(\hat{\Sigma}) > 1 + \frac{\lambda}{2}$ to distinguish the case of $H_0$ and $H_v$, with $n = \widetilde{\Omega}(k/\lambda^2)$ samples. However, this test is highly inefficient, as the best known ways for computing $\lambda_{\max}^k(\hat{\Sigma})$ take exponential time! We now turn to consider efficient ways of solving this problem.

## 2.1   Sum of Squares (Lasserre) Relaxations

Here we will only briefly introduce the basic ideas of Sum-of-Squares (Lasserre) relaxation that will be used for this paper. We refer readers to the extensive [20, 21, 27] for detailed discussions of sum of squares algorithms and proofs and their applications to algorithm design.

Let $\mathbb{R}[x]_d$ denote the set of all real polynomials of degree at most $d$ with $n$ variables $x_1, \ldots, x_n$. We start by defining the notion of *pseudo-moment* (sometimes called *pseudo-expectation* ). The intuition is that these pseudo-moments behave like the actual first $d$ moments of a real probability distribution.

**Definition 2.2** (pseudo-moment). A degree-$d$ pseudo-moments $M$ is a linear operator that maps $\mathbb{R}[x]_d$ to $\mathbb{R}$ and satisfies $M(1) = 1$ and $M(p^2(x)) \geq 0$ for all real polynomials $p(x)$ of degree at most $d/2$.

For a mutli-set $S \subset [n]$, we use $x^S$ to denote the monomial $\prod_{i \in S} x_i$. Since $M$ is a linear operator, it can be clearly described by all the values of $M$ on the monomial of degree $d$, that is, all the values of $M(x^S)$ for mutli-set $S$ of size at most $d$ uniquely determines $M$. Moreover, the nonnegativity constraint $M(p(x)^2) \geq 0$ is equivalent to the positive semidefiniteness of the matrix-form (as defined below), and therefore the set of all pseudo-moments is convex.

**Definition 2.3** (matrix-form). For an even integer $d$ and any degree-$d$ pseudo-moments $M$, we define the matrix-form of $M$ as the trivial way of viewing all the values of $M$ on monomials as a matrix: we use $\text{mat}(M)$ to denote the matrix that is indexed by multi-subset $S$ of $[n]$ with size at most $d/2$, and $\text{mat}(M)_{S,T} = M(x^S x^T)$.

Given polynomials $p(x)$ and $q_1(x), \ldots, q_m(x)$ of degree at most $d$, and a polynomial program,

$$\begin{aligned} \text{Maximize} \quad & p(x) & (2.3) \\ \text{Subject to} \quad & q_i(x) = 0, \forall i \in [m] \end{aligned}$$

We can write a sum of squares based relaxation in the following way: Instead of searching over $x \in \mathbb{R}^n$, we search over all the possible "pseudo-moments" $M$ of a hypothetical distribution over solutions $x$, that satisfy the constraints above. The key of the relaxation is to consider only moments up to degree $d$. Concretely, we have the following semidefinite program in roughly $n^d$ variables.

$$\begin{aligned} \text{Variables} \quad & M(x^S) & \forall S : |S| \leq d \\ \text{Maximize} \quad & M(p(x)) & (2.4) \\ \text{Subject to} \quad & M(q_i(x)x^K) = 0 & \forall i, K : |K| + \deg(q_i) \leq d \\ & \text{mat}(M) \succeq 0 \end{aligned}$$

Note that (2.4) is a valid relaxation because for any solution $x_*$ of (2.3), if we define $M(x^S)$ to be $M(x^S) = x_*^S$, then $M$ satisfies all the constraints and the objective value is $p(x_*)$. Therefore it is guaranteed that the optimal value of (2.4) is always larger than that of (2.3).

Finally, the key point is that this program can be solved efficiently, in polynomial time in its size, namely in time $n^{O(d)}$. As $d$ grows, the constraints added make the "pseudo-distribution" defined by the moments closer and closer to an actual distribution, thus providing a tighter relaxation, at the cost of a larger running time to solve it. In the next section we apply this relaxation to the Sparse PCA problem and state our results.

# 3   Main Results

To exploit the sum of squares relaxation framework as described in Section 2.1], we first convert the statistically optimal estimator/detector (2.1) into the "polynomial" program version below.

$$\text{Maximize} \langle \hat{\Sigma}, xx^T \rangle \tag{3.1}$$

$$\text{subject to} \|x\|_2^2 = k, \text{and } x_i^3 = x_i, \forall i \in [p] \tag{3.2\& 3.3}$$

$$|x|_1 \leq k \tag{3.4}$$

The non-convex sparsity constraint (2.2) is replaced by the polynomial constraint (3.3), which ensures that any solution vector $x$ has entries in $\{0, \pm 1\}$, and so together with the constraint (3.2) guarantees that it has precisely $k$ non-zero $\pm 1$ entries. The constraint (.3.3) implies other natural constraints that one may add to the program in order to make it stronger: for example, the upper bound on each entry $x_i$, the lower bound on the non-zero entries of $x_i$, and the constraint $\|x\|^4 \geq k$ which is used as a surrogate for $k$-sparse vectors in [25, 24]. Note that we also added an $\ell_1$ sparsity constraint (3.4) (which is convex) as is often used in practice and makes our lower bound even stronger. Of course, it is formally implied by the other constraints, but not in low-degree SoS.

Now we are ready to apply the sum-of-squares relaxation scheme described in Section 2.1) to the polynomial program above as . For degree-4 relaxation we obtain the following semidefinite program $\text{SoS}_4(\hat{\Sigma})$, which we view as an algorithm for both detection and estimation problems. Note

that the same objective function, with only the three constraints (C1&2), (C6) gives the degree-2 relaxation, which is precisely the standard SDP relaxation of Sparse PCA studied in [42, 41, 45]. So clearly $\text{SoS}_4(\hat{\Sigma})$ subsumes the SDP relaxation.

---

**Algorithm 1** $\text{SoS}_4(\hat{\Sigma})$: Degree-4 Sum of Squares Relaxation

---

Solve the following SDP and obtain optimal objective value $\text{SoS}_4(\hat{\Sigma})$ and maximizer $M^*$.

**Variables:** $M(S)$, for all mutli-sets $S$ of size at most 4.

$$\text{SoS}_4(\hat{\Sigma}) = \max \quad \sum_{i,j} M(x_i x_j) \hat{\Sigma}_{ij} \tag{Obj}$$

$$\text{subject to} \quad \sum_{i \in [p]} M(x_i^2) = k \quad \text{and} \quad \sum_{i,j \in [p]} |M(x_i x_j)| \le k^2 \tag{C1\&2}$$

$$M(x_i^3 x_j) = M(x_i x_j), \text{ and } \sum_{\ell \in [p]} M(x_\ell^2 x_i x_j) = k M(x_i x_j), \forall i, j \in [p] \tag{C4}$$

$$\sum_{i,j,s,t \in [p]} |M(x_i x_j x_s x_t)| \le k^4 \quad \text{and} \quad M \succeq 0 \tag{C5\&6}$$

**Output:** 1. For detection problem : output $H_v$ if $\text{SoS}_4(\hat{\Sigma}) > (1 + \frac{1}{2}\lambda)k$, $H_0$ otherwise
2. For estimation problem: output $M_2^* = (M^*(x_i x_j))_{i,j \in [p]}$

---

Before stating the lower bounds for both detection and estimation in the next two subsections, we comment on the choices made for the outputs of the algorithm in both, as clearly other choices can be made that would be interesting to investigate. For *detection*, we pick the natural threshold $(1 + \frac{1}{2}\lambda)k$ from the statistically optimal detection algorithm of Section 2. Our lower bound of the objective under $H_0$ is actually a large constant multiple of $\lambda k$, so we could have taken a higher threshold. To analyze even higher ones would require analyzing the behavior of $\text{SoS}_4$ under the (planted) alternative distribution $H_v$. For *estimation* we output the maximizer $M_2^*$ of the objective function, and prove that it is not too correlated with the rank-1 matrix $vv^T$ in the planted distribution $H_v$. This suggest, but does not prove, that the leading eigenvector of $M_2^*$ (which is a natural estimator for $v$) is not too correlated with $v$. We finally note that Rigollet's efficient reduction from detection to estimation is not in the SoS framework, and so our detection lower bound does not automatically imply the one for estimation.

For the detection problem, we prove that $\text{SoS}_4(\hat{\Sigma})$ gives a large objective on null hypothesis $H_0$.

**Theorem 3.1.** There exists absolute constant $C$ and $r$ such that for $1 \le \lambda < \min\{k^{1/4}, \sqrt{n}\}$ and any $p \ge C\lambda n$, $k \ge C\lambda^{7/6}\sqrt{n} \log^r p$, the following holds. When the data $X$ is drawn from the null hypothesis $H_0$, then with high probability $(1 - p^{-10})$, the objective value of degree-4 sum of squares relaxation $\text{SoS}_4(\hat{\Sigma})$ is at least $10\lambda k$. Consequently, Algorithm 1 can't solve the detection problem.

To parse the theorem and to understand its consequence, consider first the case when $\lambda$ is a constant (which is also arguably the most interesting regime). Then the theorem says that when we have only $n \ll k^2$ samples, degree-4 SoS relaxation $\text{SoS}_4$ still overfits heavily to the randomness of the data $X$ under the null hypothesis $H_0$. Therefore, using $\text{SoS}_4(\hat{\Sigma}) > (1 + \frac{\lambda}{2})k$ (or even $10\lambda k$) as a threshold will fail with high probability to distinguish $H_0$ and $H_v$.

We note that for constant $\lambda$ our result is essentially tight in terms of the dependencies between $n, k, p$. The condition $p = \widetilde{\Omega}(n)$ is necessary since otherwise when $p = o(n)$, even without the sum of squares relaxation, the objective value is controlled by $(1 + o(1))k$ since $\hat{\Sigma}$ has maximum eigenvalue $1 + o(1)$ in this regime. Furthermore, as mentioned in the introduction, $k \ge \widetilde{\Omega}(\sqrt{n})$ is also necessary (up to poly-logarithmic factors), since when $n \gg k^2$, a simple diagonal thresholding algorithm works for this simple single-spike model.

When $\lambda$ is not considered as a constant, the dependence of the lower bound on $\lambda$ is not optimal, but close. Ideally one could expect that as long as $k \gg \lambda\sqrt{n}$, and $p \ge \lambda n$, the objective value on the null hypothesis is at least $\Omega(\lambda k)$. Tightening the $\lambda^{1/6}$ slack, and possibly extending the range of

$\lambda$ are left to future study. Finally, we note that he result can be extended to a lower bound for the estimation problem, which is presented in the supplementary material.

## Footnotes

[1]We treat $\lambda$ as a constant so that we omit the dependence on it for simplicity throughout the introduction section

[2]An average case version of the Clique problem in which the input is a random graph in which a much larger than expected clique is planted.

[3]Or we assume that they go to infinity as typically done in statistics.

[4]A real random variable $X$ is subgaussian with variance proxy $\sigma^2$ if it has similar tail behavior as gaussian distribution with variance $\sigma^2$. More formally, if for any $t \in \mathbb{R}$, $\mathbb{E}[\exp(tX)] \leq \exp(t^2\sigma^2/2)$

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
