[Supplementary Material]

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

. Then we state our main results in Section 3. In Section 4, we design the pseudo-moments and state their properties and then in Section 5 we prove our main theorems using these moments. Section 6 and 7 contain the analysis of the moments. Section 8 lists the tools that we heavily used for proving concentration inequalities in the analysis. Finally we conclude with a discussion of further directions of study in Section 9.

## 2 Formal description of the model and problem

**Notation:** We use $\|\cdot\|$ to denote the euclidean norm of a vector and spectral norm of a matrix, $\|\cdot\|_q$ to denote the $q$-norm of a vector, and $|\cdot|_0$ is the number of nonzero entries of a vector. We use $[m]$ to denote the set of integers $\{1, \ldots, m\}$.

We write $M \succeq 0$ if $M$ is a positive semidefinite matrix.

$\mathbb{R}_n[x]_d$ is used to denote the set of real polynomials with $n$ variables and degree at most $d$. We will drop the subscript $n$ when it is clear from context. We will assume that $n, k, p$ are all sufficiently large[3], and that $n \leq p$.

Throughout this paper, by "with high probability some event happens", we mean the failure probability is bounded by $p^{-c}$ for every constant $c$, as $p$ tends to infinity.

We use the asymptotic notation $\widetilde{O}(\cdot)$ and $\widetilde{\Omega}(\cdot)$ to hide the logarithmic dependency (in $p$). That is, $m \leq \widetilde{O}(f(n, p, k))$ means that there exists universal constant $r \geq 0$ (which is less than 3 typically in this paper) and $C$ such that $m \leq Cf(n, p, k) \log^r p$, and $m \geq \widetilde{\Omega}(f(n, p, k))$ means that there exist constants $r$ and $c$ such that $m \geq cf(n, p, k)/\log^r p$.

## 2.1 Sparse PCA estimation and detection problems

We will consider the simplest setting of sparse PCA, which is called single-spiked covariance model in literature [Joh01] (note that restricting to a special case makes our lower bound hold in all generalizations of this simple model). In this model, the task is to recover a single sparse vector from noisy samples as follows. The "hidden data" is an unknown $k$-sparse vector $v \in \mathbb{R}^p$ with $|v|_0 = k$ and $\|v\| = 1$. To make the task easier (and so the lower bound stronger), we even assume that $v$ has discrete entries, namely that $v_i \in \{0, \pm\frac{1}{\sqrt{k}}\}$ for all $i \in [p]$. We observe $n$ noisy samples $X^1, \ldots, X^n \in \mathbb{R}^p$ that are generated as follows. Each is independently drawn as

$$X^j = \sqrt{\lambda} g^j v + \xi^j \tag{2.1}$$

from a distribution which generalizes both Gaussian and Bernoulli noise to $v$. Namely, the $g^j$'s are i.i.d real random variable with mean 0 and variance 1, and $\xi^j$'s are i.i.d random vectors which have independent entries with mean zero and variance 1. Therefore under this model, the covariance of $X^i$ is equal to $\lambda vv^T + I$. Moreover, we assume that $g^j$ and entries of $\xi^j$ are sub-gaussian[4] with variance proxy $O(1)$. Given these samples, the *estimation* problem is to approximate the unknown sparse vector $v$.

It is also interesting to also consider the sparse component *detection* problem [BR13b, BR13a], which is the decision problem of distinguishing from random samples the following two distributions

$H_0$: data $X^j = \xi^j$ is purely random

$H_v$: data $X^j = \xi^j + \sqrt{\lambda} g^j v$ contains a hidden sparse signal with strength $\lambda$.

Rigollet [MR14] observed that a polynomial time algorithm for estimation version of sparse PCA with constant error implies that an algorithm for the detection problem with twice number of the samples. Thus, for polynomial time lower bounds, it suffices to consider the detection problem.

We will use $X$ as a shorthand for the $p \times n$ matrix $[X^1, \ldots, X^n]$. We denote the rows of $X$ as $X_1^T, \ldots, X_p^T$, therefore $X_i$'s are $n$-dimensional column vectors. The empirical covariance matrix is defined as $\hat{\Sigma} = \frac{1}{n}XX^T$.

## 2.2 Statistically optimal estimator/detector

It is well known that the following non-convex program achieves optimal statistical minimax rate for the estimation problem and the optimal sample complexity for the detection problem. Note that we scale the variables $x$ up by a factor of $\sqrt{k}$ for simplicity (the hidden vector now has entries from $\{0, \pm 1\}$).

$$\lambda_{\max}^k(\hat{\Sigma}) = \frac{1}{k} \cdot \max \qquad \langle \hat{\Sigma}, xx^T \rangle \qquad (2.2)$$

$$\text{subject to} \qquad \|x\|_2^2 = k \qquad (2.3)$$

$$\|x\|_0 = k \qquad (2.4)$$

**Proposition 2.1** ([AW09], [BR13b], [VL12] informally stated)**.** The non-convex program (2.2) statistically optimally solves the sparse PCA problem when $n \geq Ck/\lambda^2 \log p$ for some sufficiently large $C$. Namely, the following hold with high probability. If $X$ is generated from $H_v$, then optimal solution $x_{\text{opt}}$ of program (2.2) satisfies $\|\frac{1}{k} \cdot x_{\text{opt}} x_{\text{opt}}^T - vv^T\| \leq \frac{1}{3}$, and the objective value $\lambda_{\max}^k(\hat{\Sigma})$ is at least $1 + \frac{2\lambda}{3}$. On the other hand, if $X$ is generated from null hypothesis $H_0$, then $\lambda_{\max}^k(\hat{\Sigma})$ is at most $1 + \frac{\lambda}{3}$ .

Therefore, for the detection problem, once can simply use the test $\lambda_{\max}^k(\hat{\Sigma}) > 1 + \frac{\lambda}{2}$ to distinguish the case of $H_0$ and $H_v$, with $n = \widetilde{\Omega}(k/\lambda^2)$ samples. However, this test is highly inefficient, as the best known ways for computing $\lambda_{\max}^k(\hat{\Sigma})$ take exponential time! We now turn to consider efficient ways of solving this problem.

## 2.3 Sum of Squares (Lasserre) Relaxations

Here we will only briefly introduce the basic ideas of Sum-of-Squares (Lasserre) relaxation that will be used for this paper. We refer readers to the extensive [Las15, Lau09, BS14] for detailed discussions of sum of squares algorithms and proofs and their applications to algorithm design.

Let $\mathbb{R}[x]_d$ denote the set of all real polynomials of degree at most $d$ with $n$ variables $x_1, \ldots, x_n$. We start by defining the notion of *pseudo-moment* (sometimes called *pseudo-expectation* ). The intuition is that these pseudo-moments behave like the actual first $d$ moments of a real probability distribution.

**Definition 2.2** (pseudo-moment)**.** A degree-$d$ pseudo-moments $M$ is a linear operator that maps $\mathbb{R}[x]_d$ to $\mathbb{R}$ and satisfies $M(1) = 1$ and $M(p^2(x)) \geq 0$ for all real polynomials $p(x)$ of degree at most $d/2$.

For a mutli-set $S \subset [n]$, we use $x^S$ to denote the monomial $\prod_{i \in S} x_i$. Since $M$ is a linear operator, it can be clearly described by all the values of $M$ on the monomial of degree $d$, that is, all the values of $M(x^S)$ for mutli-set $S$ of size at most $d$ uniquely determines $M$. Moreover, the nonnegativity constraint $M(p(x)^2) \geq 0$ is equivalent to the positive semidefiniteness of the matrix-form (as defined below), and therefore the set of all pseudo-moments is convex.

**Definition 2.3** (matrix-form)**.** For an even integer $d$ and any degree-$d$ pseudo-moments $M$, we define the matrix-form of $M$ as the trivial way of viewing all the values of $M$ on monomials as a matrix: we use $\text{mat}(M)$ to denote the matrix that is indexed by multi-subset $S$ of $[n]$ with size at most $d/2$, and $\text{mat}(M)_{S,T} = M(x^S x^T)$.

Given polynomials $p(x)$ and $q_1(x), \ldots, q_m(x)$ of degree at most $d$, and a polynomial program,

$$\text{Maximize} \quad p(x) \tag{2.5}$$
$$\text{Subject to} \quad q_i(x) = 0, \forall i \in [m]$$

We can write a sum of squares based relaxation in the following way: Instead of searching over $x \in \mathbb{R}^n$, we search over all the possible "pseudo-moments" $M$ of a hypothetical distribution over solutions $x$, that satisfy the constraints above. The key of the relaxation is to consider only moments up to degree $d$. Concretely, we have the following semidefinite program in roughly $n^d$ variables.

$$\begin{aligned}
\text{Variables} \quad & M(x^S) & \forall S : |S| \leq d \\
\text{Maximize} \quad & M(p(x)) & \\
\text{Subject to} \quad & M(q_i(x)x^K) = 0 & \forall i, K : |K| + \deg(q_i) \leq d \\
& \text{mat}(M) \succeq 0 &
\end{aligned} \tag{2.6}$$

Note that (2.6) is a valid relaxation because for any solution $x_*$ of (2.5), if we define $M(x^S)$ to be $M(x^S) = x_*^S$, then $M$ satisfies all the constraints and the objective value is $p(x_*)$. Therefore it is guaranteed that the optimal value of (2.6) is always larger than that of (2.5).

Finally, the key point is that this program can be solved efficiently, in polynomial time in its size, namely in time $n^{O(d)}$. As $d$ grows, the constraints added make the "pseudo-distribution" defined by the moments closer and closer to an actual distribution, thus providing a tighter relaxation, at the cost of a larger running time to solve it.

In the next section we apply this relaxation to the Sparse PCA problem and state our results.

## 3 Main Results

To exploit the sum of squares relaxation framework as described in Section 2.3], we first convert the statistically optimal estimator/detector (2.2) into the "polynomial" program version below.

$$\begin{aligned}
\text{Maximize} \quad & \langle \hat{\Sigma}, xx^T \rangle & (3.1) \\
\text{subject to} \quad & \|x\|_2^2 = k & (3.2) \\
& x_i^3 = x_i, \forall i \in [p] & (3.3) \\
& |x|_1 \leq k & (3.4)
\end{aligned}$$

Note that the non-convex sparsity constraint (2.4) is replaced by the polynomial constraint 3.3, which ensures that any solution vector $x$ has entries in $\{0, \pm 1\}$, and so together with the constraint (3.2) guarantees that it has precisely $k$ non-zero entries, each of absolute value 1. Note that constraint (3.3) implies other natural constraints that one may add to the program in order to make it stronger: for example, the upper bound on each entry $x_i$, the lower bound on the non-zero entries of $x_i$, and the constraint $\|x\|^4 \geq k$ which has been used as a surrogate for $k$-sparse vectors in [BKS14, BKS15]. Note that we have also added an $\ell_1$ sparsity constraint (3.4) (which can be easily made into a polynomial constraint) as is often used in practice and makes our lower bound even stronger. Of course, it is formally implied by the other constraints, but not in low-degree SoS.

Now we are ready to apply the sum-of-squares relaxation scheme described in Section 2.3) to the polynomial program above as . For degree-4 relaxation we obtain the following semidefinite

program $\text{SoS}_4(\hat{\Sigma})$, which we view as an algorithm for both detection and estimation problems. Note that the same objective function, with only the three constraints (C1), (C2), (C6) gives the degree-2 relaxation, which is precisely the standard SDP relaxation of Sparse PCA studied in [AW09, BR13b, KNV15]. So clearly $\text{SoS}_4(\hat{\Sigma})$ subsumes the SDP relaxation.

---

**Algorithm 1** $\text{SoS}_4(\hat{\Sigma})$: Degree-4 Sum of Squares Relaxation

---

**Input:** $\hat{\Sigma} = \frac{1}{n} X X^T$ where $X = \left[ X^1, \ldots, X^n \right] \in \mathbb{R}^{p \times n}$

Solve the following semidefinite programming and obtain optimal objective value $\text{SoS}_4(\hat{\Sigma})$ and maximizer $M^*$.

**Variables:** $M(S)$, for all mutli-sets $S$ of size at most 4.

$$\text{SoS}_4(\hat{\Sigma}) = \max \quad \sum_{i,j} M(x_i x_j)\hat{\Sigma}_{ij} \tag{Obj}$$

$$\text{subject to} \quad \sum_{i \in [p]} M(x_i^2) = k \tag{C1}$$

$$\sum_{i,j \in [p]} |M(x_i x_j)| \leq k^2 \tag{C2}$$

$$M(x_i^3 x_j) = M(x_i x_j), \quad \forall i, j \in [p] \tag{C3}$$

$$\sum_{i \in [p]} M(x_i^2 x_s x_t) = k \cdot M(x_s x_t), \quad \forall s, t \in [p] \tag{C4}$$

$$\sum_{i,j,s,t \in [p]} |M(x_i x_j x_s x_t)| \leq k^4 \tag{C5}$$

$$M \succeq 0 \tag{C6}$$

**Output:** 1. For detection problem : output $H_v$ if $\text{SoS}_4(\hat{\Sigma}) > (1 + \frac{1}{2}\lambda)k$, $H_0$ otherwise
2. For estimation problem: output $M_2^* = (M^*(x_i x_j))_{i,j \in [p]}$

---

Before stating the lower bounds for both detection and estimation in the next two subsections, we comment on the choices made for the outputs of the algorithm in both, as clearly other choices can be made that would be interesting to investigate. For *detection*, we pick the natural threshold $(1 + \frac{1}{2}\lambda)k$ from the statistically optimal detection algorithm of Section 2.2. Our lower bound of the objective under $H_0$ is actually a large constant multiple of $\lambda k$, so we could have taken a higher threshold. To analyze even higher ones would require analyzing the behavior of $\text{SoS}_4$ under the (planted) alternative distribution $H_v$. For *estimation* we output the maximizer $M_2^*$ of the objective function, and prove that it is not too correlated with the rank-1 matrix $vv^T$ in the planted distribution $H_v$. This suggest, but does not prove, that the leading eigenvector of $M_2^*$ (which is a natural estimator for $v$) is not too correlated with $v$. We finally note that Rigollet's efficient reduction from detection to estimation is not in the SoS framework, and so our detection lower bound does not automatically imply the one for estimation.

## 3.1  Lower bounds for detection problem

For the detection problem, we prove that $\mathrm{SoS}_4(\hat{\Sigma})$ gives a large objective value on null hypothesis $H_0$.

**Theorem 3.1.** There exists absolute constant $C$ and $r$ such that for $1 \leq \lambda < \min\{k^{1/4}, \sqrt{n}\}$ and any $p \geq C\lambda n$, $k \geq C\lambda^{7/6}\sqrt{n}\log^r p$, the following holds. When the data $X$ is drawn from the null hypothesis $H_0$, then with high probability $(1 - p^{-10})$, the objective value of degree-4 sum of squares relaxation $\mathrm{SoS}_4(\hat{\Sigma})$ is at least $10\lambda k$. Consequently, Algorithm 1 can't solve the detection problem.

To parse the theorem and to understand its consequence, consider first the case when $\lambda$ is a constant (which is also arguably the most interesting regime). Then the theorem says that when we have only $n \ll k^2$ samples, degree-4 SoS relaxation $\mathrm{SoS}_4$ still overfits heavily to the randomness of the data $X$ under the null hypothesis $H_0$. Therefore, using $\mathrm{SoS}_4(\hat{\Sigma}) > (1 + \frac{\lambda}{2})k$ (or even $10\lambda k$) as a threshold will fail with high probability to distinguish $H_0$ and $H_v$.

We note that for constant $\lambda$ our result is essentially tight in terms of the dependencies between $n, k, p$. The condition $p = \widetilde{\Omega}(n)$ is necessary since otherwise when $p = o(n)$, even without the sum of squares relaxation, the objective value is controlled by $(1 + o(1))k$ since $\hat{\Sigma}$ has maximum eigenvalue $1 + o(1)$ in this regime. Furthermore, as mentioned in the introduction, $k \geq \widetilde{\Omega}(\sqrt{n})$ is also necessary (up to poly-logarithmic factors), since when $n \gg k^2$, a simple diagonal thresholding algorithm works for this simple single-spike model.

When $\lambda$ is not considered as a constant, the dependence of the lower bound on $\lambda$ is not optimal, but close. Ideally one could expect that as long as $k \gg \lambda\sqrt{n}$, and $p \geq \lambda n$, the objective value on the null hypothesis is at least $\Omega(\lambda k)$. Tightening the $\lambda^{1/6}$ slack, and possibly extending the range of $\lambda$ are left to future study.

## 3.2  Lower bounds for the estimation problem

For estimation problem, we prove that $M_2^*$ output by Algorithm 1 is not too correlated with the desired rank-1 matrix $vv^T$.

**Theorem 3.2.** For any constant $B$ there exists absolute constants $C$ and $r$ such that for $\lambda \leq B/2$, $Bn \geq p \geq 2\lambda n$ and $o(p) \geq k \geq C\sqrt{n}\log^r p$, suppose the data $X$ is drawn hypothesis $H_v$ (model (2.1)), then with high probability $(1 - p^{-10})$ over the randomness of the data, Algorithm 1 will output $M_2^*$ such that $\|\frac{1}{k} \cdot M_2^* - vv^T\| \geq 1/5$.

We observe that the result is of the same nature (and arguably near-optimal for estimation problem) as [KNV15] achieve the for degree-2 SoS relaxation. The proof follows simply from combining our detection lower bound Theorem 3.1 and arguments similar to [KNV15]. Finally we address a threshold-like behavior of the estimation error. Note that while our Theorem proves that $n = \widetilde{\Omega}(k^2)$ samples is necessary for efficient algorithms to get even constant estimation error, it is known [YZ13, Ma13, WLL14] that slightly more samples, $n = \widetilde{O}(k^2)$, can already achieve in polynomial time a much smaller (and optimal) estimation error, namely $O(\sqrt{(k\log p)/n})$.

# 4  Design of Pseudo-moments

We start with a sketch of our approach to the design of the moments $M$ at a very high level, highlighting aspects of their design which are different than in previous lower bounds. First, there

are some natural choices to make. We define the degree-2 moments $\widetilde{M}$ from the input as the empirical covariance matrix, as was done in the proof of the SDP lower bound. This already gives a large objective value (see Lemma 4.2). We also define taking odd moments (degree 1 and 3) to be 0. The difficult part is designing the degree-4 moments consistently with the constraints and $\widetilde{M}$. We do this in stages, first approximating the constraints (indeed even $\widetilde{M}$ only approximately satisfies, in a way we will specify in Section 4.1, constraints (C1) and (C2)) and later in Section 4.2 correcting the moments to satisfy the constraints precisely. Moreover, we separately use different 4-moments for different constraints and then combine them, as follows. We define two different degree-4 PSD moments $P$ and $Q$ such that (with high probability) $P$ *almost* satisfies constraints (C3), (C5) and (C6), and *negligible* for constraint (C4) (see Lemma 4.4), whereas $Q$ *almost* satisfies constraints (C5), (C4) and (C6), and *negligible* for (C3) (Lemma 4.5). Therefore taking the sum $P + Q$ will *almost* satisfy all the constraints (Lemma 4.6), which completes the design of the approximate moments. Finally we "locally" adjust $P + Q$ so that the resulting moments $M$ exactly satisfy all the constraints (Theorem 4.7), and remain PSD with high probability.

All moments will be defined from the data matrix $X$, to which we first apply a simple pre-processing step: we scale all its rows to have square norm $n$ (around which they are concentrated). We abuse notation and call the scaled matrix $X$ as well. Note that when the noise model in the null hypothesis $H_0$ is Bernoulli, namely the entries of $X$ are chosen as unbiased independent $\pm 1$ variables, the rows are automatically scaled, which motivates our abuse of notation. We suggest that the reader thinks of this distribution, even though the proof works for a much wider class of distributions.

The properties above of our moments will be proved under the assumption that the scaled matrix $X$ satisfies the "pseudo-randomness" condition below. This set-up allows us to encapsulate what we really need the data to satisfy, and thus prove our lower bound not only for Gaussian or Bernoulli noise, but actually for a larger family containing both. Namely, we later prove in Section 7, via a series of concentration inequalities, that when data is drawn from null hypothesis $H_0$, its scaling $X$ satisfies the pseudorandomness condition with very high probability under all these noise models. Note that this condition is actually a sequence of statements about deviation from the mean of various polynomials in the data - these will become natural once we define our moments.

**Condition 4.1** (Pseudorandomness Condition)**.** Our constructions of the moments will only require

the following pseudorandom conditions about the (scaled) data matrix $X$,

$$\|X_i\|^2 = n \quad \forall i \in [p] \tag{P1}$$

$$|\langle X_i, X_j \rangle| \leq \widetilde{O}(\sqrt{n}) \qquad \forall i \neq j \tag{P2}$$

$$\left| \sum_{\ell \in [p] \setminus i \cup j} \langle X_i, X_\ell \rangle^3 \langle X_j, X_\ell \rangle \right| \leq \widetilde{O}(n^{1.5}p), \qquad \forall i \neq j \tag{P3}$$

$$\left| \sum_{\ell \in [p]} \langle X_i, X_\ell \rangle \langle X_j, X_\ell \rangle \langle X_s, X_\ell \rangle \langle X_t, X_\ell \rangle \right| \leq \widetilde{O}(n^2 p^{.5}) \qquad \forall \text{ disctinct } i, j, s, t \tag{P4}$$

$$\left| \sum_{i \in [p]} \langle X_i, X_s \rangle \langle X_i, X_t \rangle \right| \leq \widetilde{O}(n^{.5}p) \qquad \forall \text{ distinct } s, t \tag{P5}$$

$$\sum_{i, \ell \in [p]} \langle X_i, X_\ell \rangle^2 \langle X_s, X_\ell \rangle \langle X_t, X_\ell \rangle \leq \widetilde{O}(n^{1.5}p^2), \qquad \forall \text{ distinct } s, t \tag{P6}$$

$$\|XX^T\|_F^2 \geq (1 - o(1))np^2 \tag{P7}$$

## 4.1   Approximate Pseudo-moments

In this section, we design a pseudo-moments $\widetilde{M}$ that approximately satisfies the all the constraints. Then in the next subsection we will locally adjust it to obtain one that exactly satisfies all of the constraints.

We begin by designing a (partial) degree-2 moments that gives large objective value, which will be later used for the degree-4 moments. The design is essentially the same as [KNV15] though we only work with null hypothesis for now. For the purpose of this section, we suggest the reader to think of $X$ as having uniform $\{\pm 1\}$ entires for simplicity, though as we will see later, we assume that $X$ satisfies certain pseudorandomness condition which holds if $X$ is chosen from a variety of natural stochastic models (with row normalization). We define $\widetilde{M} : \mathbb{R}[x]_2 \to \mathbb{R}$ as follows:

$$\begin{aligned}
\widetilde{M}(x_i x_j) &\triangleq \frac{\gamma k n}{p^2} \hat{\Sigma}_{ij} = \frac{\gamma k}{p^2} \langle X_i, X_j \rangle \quad \forall i, j \in [p] \tag{4.1} \\
\widetilde{M}(x_i) &\triangleq 0 \quad \forall i \in [p] \\
\widetilde{M}(1) &\triangleq 1
\end{aligned}$$

where $\gamma$ is a constant that to be tuned later according to the signal strength $\lambda$. Note that by design $\text{mat}(\widetilde{M})$ is a PSD matrix. We can check straightforwardly that $\widetilde{M}$ satisfies constraint (C2) and gives a large objective value (Obj).

**Lemma 4.2.** There exists constant $C$ such that for $p \geq \gamma n$ and $k \geq C\gamma\sqrt{n}\log p$, suppose $X$ satisfies Condition 4.1, then $\widetilde{M}$ is a valid degree-2 pseudo-moments and satisfies the sparsity constraint (C2).

$$\sum_{i,j \in [p]} |\widetilde{M}(x_i x_j)| \leq k^2/2 \tag{4.2}$$

and has objective value

$$\sum_{i,j \in [p]} \widetilde{M}(x_i x_j) \hat{\Sigma}_{ij} \geq (1 - o(1))\gamma k$$

Moreover, we also have $\widetilde{M}(x_i^2) = \frac{\gamma k n}{p^2} \le \frac{k}{p}$, and $\widetilde{M}(x_i x_j) \le \widetilde{O}(\frac{\gamma k \sqrt{n}}{p^2})$.

*Proof.* The proof follows simple calculation and concentration inequality. Since $\|X_i\|^2 = n$ for all $i$ and with high probability over the randomness of $X$, for all $i \ne j$, $|\langle X_i, X_j \rangle| \le \widetilde{O}(\sqrt{n})$, we obtain that $\widetilde{M}(x_i^2) = \frac{\gamma k n}{p^2} \le \frac{k}{p}$, and $\widetilde{M}(x_i x_j) \le \widetilde{O}(\frac{\gamma k \sqrt{n}}{p^2})$. Then to verify equation (4.2), we have

$$\sum_{i,j \in [p]} |\widetilde{M}(x_i x_j)| \le \sum_i |M(x_i^2)| + \sum_{i \ne j} |M(x_i x_j)| \le k + \widetilde{O}(\gamma k \sqrt{n}) \le k^2/2$$

when $k \gg \gamma \sqrt{n}$. Finally, we can verify the objective value is large

$$\sum_{i,j \in [p]} \widetilde{M}(x_i x_j) \hat{\Sigma}_{ij} = \frac{\gamma k}{p^2 n} \sum_{i,j} \langle X_i, X_j \rangle^2 = \frac{\gamma k}{p^2 n} \|XX^T\|_F^2 \ge (1 - o(1))\gamma k$$

where we use the fact that $\|XX^T\|_F^2 \ge (1 - o(1))p^2 n$ (see property (P7) in Condition 4.1). $\qquad \square$

Note that $\widetilde{M}$ doesn't satisfies constraint (C1) exactly. However, we could simply fix this by defining $M'(x_i x_j) = \widetilde{M}(x_i x_j)$ for all $i \ne j$ and $M'(x_i^2) = k/p$. However, note that we will use a perturbation of $M'$ in our final design in Section 4.2 so that it is consistent with the degree-4 moments.

**Corollary 4.3.** There exists absolute constant $C$ such that for $p \ge \gamma n$ and $k \ge C\gamma \sqrt{n} \log p$, there exists a degree-2 pseudo-moments $M'$ that satisfies constraints (C1), (C2) and give objective value at least $(1 - o(1))\gamma k$.

Now we define a degree-4 pseudo-moment that approximately satisfies all the constraints in $\mathrm{SoS}_4(\hat{\Sigma})$ and give a large objective value. We keep the current (approximate) design $\widetilde{M}$ for degree-2 moments, since the degree-2 moments defined in previous section seems to be nearly optimal and enjoys many good properties. Then we define $\widetilde{M}(S) = 0$ for any multi-set $S$ of size 3, because apparently degree-3 moments don't play any role the semidefinite relaxation.

The main difficulty is to define $\widetilde{M}(S)$ for $S$ of size 4. Here we have three constraints (C3), (C4), and (C5), and the PSDness constraint that implicitly compete with each other. We took the following approach. We let $\widetilde{M}$ be a sum of two matrices matrix $P$ and $Q$. We ensure that $P$ "almost" (as will be specified later) satisfies (C3) and (C5), and is negligible for constraints C4. In turn $Q$ is negligible for constraints (C3) and "almost" satisfies constraint (C4) and (C5). Therefore $P + Q$ will "almost" satisfies constraints (C3)) and (C4)), and satisfy the sparsity constraint (C5). Moreover, $P$ and $Q$ will be PSD by definition. Concretely, we define

$$\widetilde{M}(i, j, s, t) \triangleq P(i, j, s, t) + Q(i, j, s, t)$$

where $P$ and $Q$ are defined as

$$P(i, j, s, t) = \frac{\gamma k}{p^2 n^3} \sum_{\ell \in [p]} \langle X_i, X_\ell \rangle \langle X_j, X_\ell \rangle \langle X_s, X_\ell \rangle \langle X_t, X_\ell \rangle$$

$$Q(i, j, s, t) = \frac{\gamma k^2}{p^3 n} \left( \langle X_s, X_t \rangle \langle X_i, X_j \rangle + \langle X_i, X_t \rangle \langle X_s, X_j \rangle + \langle X_j, X_t \rangle \langle X_i, X_s \rangle \right)$$

We note that $P$ and $Q$ are well defined pseudo-moments because they are invariant to the permutation of indices and naturally PSD. To see the PSDness, note that $P$ is a sum of $p$ rank-1 PSD matrices. Moreover, $Q$ is also PSD: the part that correpsonds to $\langle X_s, X_t\rangle\langle X_i, X_j\rangle$ is simply a rank-1 PSD matrix; $\langle X_i, X_t\rangle\langle X_s, X_j\rangle$ can be written as $\langle X_t \otimes X_s, X_i \otimes X_j\rangle$ and therefore it also contributes a PSD matrix to $Q$. Similarly, $\langle X_j, X_t\rangle\langle X_s, X_i\rangle$ can be written as $\langle X_s \otimes X_t, X_i \otimes X_j\rangle$, and it also contributes a PSD matrix.

In the next two lemmas (one for $P$ and one for $Q$), we formalize the intuition above by showing that, the deviation from $P$ and $Q$ exactly satisfying the constraints is captured by error matrices $\mathcal{E}, \mathcal{F}, \mathcal{G}$. We bound the magnitude of these error matrices and establish the PSDness of some of them so that later we can fix them for the exact satisfaction of the constraints.

**Lemma 4.4.** There exists some absolute constant $C$ and $r$ such that for $1 \leq \gamma \leq \min\{k^{1/4}, \sqrt{n}\}$, $1 \leq \gamma \leq n$, $p = 1.1\gamma n$, and $k \geq C \cdot \gamma^{7/6}\sqrt{n}\log^r p$, suppose $X$ satisfies pseudorandomness condition 4.1, then $P$ almost satisfies constraint (C3)and (C5), naturally satisfies PSD constraint (C6), and is negligible for constraint (C4) in the sense that

$$P(x_i^3 x_j) = \widetilde{M}(x_i x_j) + \mathcal{F}_{ij}, \quad \forall i, j \in [p] \tag{4.3}$$

$$\sum_i P(x_i^2 x_s x_t) = \mathcal{E}_{st} \quad \forall s, t \in [p] \tag{4.4}$$

$$\sum_{i,j,s,t} |P(x_i x_j x_s x_t)| \leq k^4/3 \tag{4.5}$$

where $\mathcal{F}$ and $\mathcal{E}$ are $p \times p$ matrices that satisfy

1. $0 \leq \mathcal{F}_{ii} \leq \widetilde{O}\left(\frac{\gamma k}{pn}\right)$, $|\mathcal{F}_{ij}| \leq \widetilde{O}\left(\frac{\gamma k}{pn^{1.5}}\right)$ for any $i$ and $j \neq i$.

2. $\mathcal{E}$ is PSD with $|\mathcal{E}_{ss}| \leq \widetilde{O}\left(\frac{\gamma k}{n}\right)$, and $|\mathcal{E}_{st}| \leq \widetilde{O}(\frac{\gamma k}{n\sqrt{n}})$ for any $s \neq t$.

**Lemma 4.5.** There exists some absolute constant $C$ and $r$ such that for $1 \leq \gamma \leq n$, $p = 1.1\gamma n$ and $k \geq C \cdot \gamma\sqrt{n}\log^r p$, suppose $X$ satisfies pseudorandomness condition 4.1, then $Q$ is negligible for constraint (C3)) and almost satisfies constraint (C4) and (C5) in the sense that,

$$Q(x_i^3 x_j) = \frac{3k}{p}\widetilde{M}(x_i x_j) \qquad \forall i, j \tag{4.6}$$

$$\sum_i Q(x_i^2 x_s x_t) = k\widetilde{M}(x_s x_t) + \mathcal{G}_{st}, \quad \forall s, t \tag{4.7}$$

$$\sum_{i,j,s,t} |Q(x_i x_j x_s x_t)| \leq k^4/3 \tag{4.8}$$

where $\mathcal{G}$ is a $p \times p$ PSD matrix $|\mathcal{G}_{ss}| \leq \widetilde{O}\left(\frac{\gamma k^2}{p^2}\right)$ and $|\mathcal{G}_{st}| \leq \widetilde{O}\left(\frac{\gamma k^2}{p^2\sqrt{n}}\right)$ for any $s \neq t$.

Now we are ready to prove that $\widetilde{M} = P + Q$ almost satisfies all other constraints (C1)- (C6) approximately.

**Lemma 4.6.** Define $\widetilde{M}(x_i x_j x_s x_t) = P(x_i x_j x_s x_t) + Q(x_i x_j x_s x_t)$ for all $i, j, s, t \in [p]$, then we have under the condition of Lemma 4.4,

$$\widetilde{M}(x_i^3 x_j) = \widetilde{M}(x_i x_j) + \mathcal{F}'_{ij} \tag{4.9}$$

$$\sum_{i \in [p]} \widetilde{M}(x_i^2 x_s x_t) = k\widetilde{M}(x_s x_t) + \mathcal{E}'_{st} \quad \forall s, t \tag{4.10}$$

$$\sum_{i,j,s,t} |\widetilde{M}(x_i x_j x_s x_t)| \le 2k^4/3 \tag{4.11}$$

where $\mathcal{F}'$ and $\mathcal{E}'$ are $p \times p$ matrices that satisfy

1. $|\mathcal{F}'_{ii}| \le \widetilde{O}\left(\frac{\gamma k^2 n}{p^3}\right)$ and $|\mathcal{F}'_{ij}| \le \widetilde{O}\left(\frac{\gamma k^2 \sqrt{n}}{p^3}\right)$ for all $i \ne j$.

2. $\mathcal{E}'$ is a PSD matrix with $\mathcal{E}'_{ss} \le \widetilde{O}\left(\frac{\gamma k}{n}\right)$ and $|\mathcal{E}'_{st}| \le \widetilde{O}(\frac{\gamma k}{n\sqrt{n}})$ for $s \ne t$.

*Proof of Lemma 4.6 using Lemma 4.4 and Lemma 4.5.* Note that by definition of $\widetilde{M}$ and Lemma 4.4 and Lemma 4.5, we have $\mathcal{F}'_{ij} = \mathcal{F}_{ij} + \frac{3k}{p}\widetilde{M}(x_i x_j)$ and $\mathcal{E}' = \mathcal{E} + \mathcal{G}$. The bound for $\mathcal{F}'$ follows the bound for $\mathcal{F}$ and the facts that $\widetilde{M}(x_i^2) = \frac{\gamma k n}{p^2}$ and $|\widetilde{M}(x_i x_j)| \le \widetilde{O}\left(\frac{\gamma k \sqrt{n}}{p^2}\right)$. The PSDness of $\mathcal{E}'$ and the bounds for it follows straightforwardly from those of $\mathcal{E}$ and $\mathcal{G}$. Equation (4.11) follows equation (4.4) and equation (4.8). $\qquad\square$

## 4.2 Exact Pseudo-moments

Note that $\widetilde{M}$ only satisfies the constraints approximately up to some additive errors (which are carefully bounded for the purpose of the next theorem). We fix this issue by defining the actual pseudo-moments $M$ based (on a carefully chosen) local adjustment of $\widetilde{M}$. Concretely, we define $M(1) = 1$ and for all add degree monomial $x^\alpha$, $M(x^\alpha) = 0$. For distinct $i, j, s, t$, we define $M(x_i x_j x_s x_t) \triangleq \widetilde{M}(x_i x_j x_s x_t)$ and $M(x_i^2 x_s x_t) \triangleq \widetilde{M}(x_i^2 x_s x_t)$. For distinct $s, t$, we define

$$M(x_s^3 x_t) = M(x_s x_t) \triangleq \widetilde{M}(x_s x_t) + \frac{1}{k-2}\left(\mathcal{E}'_{st} - 2\mathcal{F}'_{st}\right) \tag{4.12}$$

and $M(x_s^2 x_t^2) \triangleq \widetilde{M}(x_s^2 x_s^2) + \delta$ where $\delta$ a constant (will be proved to be nonnegative) such that

$$\sum_{i \ne j} M(x_s^2 x_t^2) = \sum_{s \ne t}\left(\widetilde{M}(x_s^2 x_t^2) + \delta\right) = k^2 - k \tag{4.13}$$

Then we define

$$M(x_i^4) = M(x_i^2) \triangleq \frac{1}{k-1}\sum_{j:j \ne i} M(x_i^2 x_j^2) \tag{4.14}$$

Therefore we can see by construction, it is almost obvious that $M$ satisfies all the linear constraints (C1), (C3), (C4) exactly. Moreover, since $\mathcal{E}'$ and $\mathcal{F}'$ are small error matrices, most of the entries $M(x_i x_j x_s x_t)$ are equal or close to $\widetilde{M}(x_i x_j x_s x_t)$. Note that $\widetilde{M}$ satisfies the rest of constraints (C2), (C5) and (C6) (even with some slackness). We will prove that the difference between $M$ and $\widetilde{M}$ is small enough so that these constraints are still satisfied by $M$.

**Theorem 4.7.** Under the condition of Lemma 4.4, suppose $X$ satisfies pseudorandomness condition 4.1, then the pseudo-moments $M$ defined above satisfies all the constraint (C1)-(C6) of the semidefinite programming and has objective value larger than $(1 - o(1))\gamma k$.

*Proof.* We prove that $M$ satisfies all the constraints in an order that is most convenient for the proof, and check the objective value at the end.

- Constraint (C3): This is satisfied by the definition of $M$.

- Constraint (C4): By the definition, we can see that $M(x_s^3 x_t)$ is also a perturbation of $\widetilde{M}(x_s^3 x_t)$:

$$M(x_s^3 x_t) = \widetilde{M}(x_s x_t) + \frac{1}{k-2}\left(\mathcal{E}'_{st} - 2\mathcal{F}'_{st}\right) = \widetilde{M}(x_s^3 x_t) + \frac{\mathcal{E}'_{st}}{k-2} - \frac{k}{k-2}\mathcal{F}'_{st} \qquad (4.15)$$

It follows that for $s \neq t$,

$$
\begin{aligned}
\sum_{i \in [p]} M(x_i^2 x_s x_t) &= 2M(x_s x_t) + \sum_{i \in [p]\setminus\{s,t\}} \widetilde{M}(x_i^2 x_s x_t) \\
&= 2\widetilde{M}(x_s x_t) + \frac{2}{k-2}\left(\mathcal{E}'_{st} - 2\mathcal{F}'_{st}\right) + \sum_{i \in [p]\setminus\{s,t\}} \widetilde{M}(x_i^2 x_s x_t) \\
&= \widetilde{M}(x_s^3 x_t) + \widetilde{M}(x_s x_t^3) - 2\mathcal{F}'_{st} + \frac{2}{k-2}\left(\mathcal{E}'_{st} - 2\mathcal{F}'_{st}\right) + \sum_{i \in [p]\setminus\{s,t\}} \widetilde{M}(x_i^2 x_s x_t) \\
&= k\widetilde{M}(x_s x_t) + \mathcal{E}'_{st} - 2\mathcal{F}'_{st} + \frac{2}{k-2}\left(\mathcal{E}'_{st} - 2\mathcal{F}'_{st}\right) \\
&= kM(x_s x_t)
\end{aligned}
$$

where the second equality uses definition (4.12) and the third uses equation (4.9), and the fourth uses (4.10) and the last equality uses the definition (4.12) again.

Moreover, for the case when $s = t$, we have that

$$
\begin{aligned}
\sum_{i \in [p]} M(x_i^2 x_s^2) &= M(x_s^4) + \sum_{i \in [p]\setminus\{s\}} M(x_i^2 x_s^2) \\
&= M(x_s^2) + (k-1)M(x_s^2) = kM(x_s^2)
\end{aligned}
$$

where we used the definition (4.14) of $M(x_s^4)$ and $M(x_s^2)$. Therefore we verified that $M$ satisfies constraint (C4).

- Constraint (C1): Using equation (4.13) and (4.14), we have

$$\sum_{i \in [p]} M(x_i^2) = \frac{1}{k-1}\sum_{i \neq j} M(x_i^2 x_j^2) = k$$

- Constraint (C6):

Next we check the PSDness of matrix $\mathrm{mat}(M)$. Note that $\mathrm{mat}(M)$ is indexed by all the mutli subset of $[p]$ of size at most 2, and it consists of 3 blocks $\mathrm{mat}(M) = \mathrm{blkdiag}(M_4, M_2, M_0)$, where

$$M_4 = (\mathrm{mat}(M)_{S,T})_{|S|=2,|T|=2}$$

$$M_2 = (\mathrm{mat}(M)_{S,T})_{|S|=1,|T|=1}$$

$$M_0 = 1$$

Therefore it suffices to check that $M_0$, $M_2$ and $M_4$ are all PSD. $M_0$ is trivially PSD. We can write $M_2$ in the following form

$$M_2 = (M(x_s x_t))_{s,t\in[p]} = \left(\widetilde{M}(x_s x_t)\right)_{s,t\in[p]} + \Delta$$

where $\Delta = M_2 - \left(\widetilde{M}(x_s x_t)\right)_{s,t\in[p]}$. By equation (4.12), we have that for $s \neq t$, $\Delta_{st} = \frac{1}{k-2}\left(\mathcal{E}'_{st} - 2\mathcal{F}'_{st}\right)$ for all $s \neq t$. Moreover, by definition of $M(x_s^2)$ and $M(x_s^2 x_t^2)$, we have that

$$
\begin{aligned}
M(x_s^2) &= \frac{1}{k-1}\sum_{s:s\neq t} M(x_s^2 x_t^2) = \frac{1}{k-1}\sum_{s:s\neq t}\left(\widetilde{M}(x_s^2 x_t^2) + \delta\right) \\
&= \frac{1}{k-1}\left(k\widetilde{M}(x_s^2) + \mathcal{E}'_{ss} - \widetilde{M}(x_i^4)\right) + \frac{p-1}{k-1}\cdot\delta \\
&= \frac{1}{k-1}\left(k\widetilde{M}(x_s^2) + \mathcal{E}'_{ss} - \widetilde{M}(x_s^2) - \mathcal{F}'_{ss}\right) + \frac{p-1}{k-1}\cdot\delta \\
&= \widetilde{M}(x_s^2) + \frac{1}{k-1}\left(\mathcal{E}'_{ss} - \mathcal{F}'_{ss}\right) + \frac{p-1}{k-1}\cdot\delta
\end{aligned}
\tag{4.16}
$$

where second line uses equation (4.10) and the third line uses (4.9), and therefore $\Delta_{ss} = \frac{p-1}{k-1}\cdot\delta + \frac{1}{k-1}\left(\mathcal{E}'_{ss} - \mathcal{F}'_{ss}\right)$. We extract the PSD matrix $\frac{1}{k-2}\cdot\mathcal{E}'$ form $\Delta$ and obtain $\Delta' = \Delta - \frac{1}{k-2}\cdot\mathcal{E}'$. Then by this definition, $\Delta'_{ss} = \frac{p-1}{k-1}\cdot\delta + \frac{1}{k-1}\left(\mathcal{E}'_{ss} - \mathcal{F}'_{ss}\right) - \frac{1}{k-2}\mathcal{E}'_{ss}$, and $\Delta'_{st} = -\frac{2}{k-2}\mathcal{F}'_{st}$. We use Gershgorin Circle Theorem to establish the PSDness of $\Delta'$. By Lemma 4.6, we have $|\mathcal{F}_{ij}| \leq \widetilde{O}\left(\frac{\gamma k^2\sqrt{n}}{p^3}\right)$ Therefore

$$\sum_{j:j\neq i}|\Delta'_{ij}| \leq p\cdot\frac{4}{k-2}\widetilde{O}\left(\frac{\gamma k^2\sqrt{n}}{p^3}\right) \leq o\left(\frac{k}{p}\right)$$

where we used the fact that $\sqrt{n}/p = o(1)$ which follows form $p = 1.1\gamma n$ and $\gamma \leq \sqrt{n}$.

Using equation (4.16) and constrain (C1) we have that

$$k = \sum_s M(x_s^2) = \sum_s \widetilde{M}(x_s^2) + \sum_s \frac{1}{k-1}\left(\mathcal{E}'_{ss} - \mathcal{F}'_{ss}\right) + \frac{p(p-1)}{k-1}\cdot\delta \tag{4.17}$$

$$\leq \frac{\gamma kn}{p} + \widetilde{O}(1) + \frac{p(p-1)}{k-1}\cdot\delta \tag{4.18}$$

It follows that $\delta \geq (1-o(1)) \cdot \frac{k(k-1)}{12p(p-1)}$. Therefore we obtain that $\Delta'_{ii} = \frac{p-1}{k-1} \cdot \delta + \frac{1}{k-1} \left( \mathcal{E}'_{ss} - \mathcal{F}'_{ss} \right) - \frac{1}{k-2} \mathcal{E}'_{ss} \geq \frac{1}{12}(1-o(1))\frac{k}{p} - \widetilde{O}(\frac{\gamma}{n}) - \widetilde{O}\left( \frac{\gamma k n}{p^3} \right) = \frac{1}{12}(1-o(1))\frac{k}{p}$. Therefore we obtain $\Delta'_{ii} \geq \sum_{j:j\neq i} |\Delta'_{ij}|$ and by Gershgorin Circle Theorem $\Delta'$ is PSD.

Now we examine $M_4$. we write $M_4$ as

$$M_4 = \mathrm{mat}(P) + \mathrm{mat}(Q) + \Gamma$$

where $\Gamma = M_4 - (\mathrm{mat}(P) + \mathrm{mat}(Q))$. One can observe that $\Gamma$ has only non-zero entries of the form

$$
\begin{aligned}
\Gamma_{ii,ii} &= M(x_i^4) - P(x_i^4) - Q(x_i^4) = M(x_i^4) - \widetilde{M}(x_i^4) = M(x_i^2) - \widetilde{M}(x_i^2) - \mathcal{F}'_{ii} \\
&= \frac{(p-1)}{k-1} \cdot \delta + \frac{1}{k-1} \mathcal{E}'_{ii} - \frac{k}{k-1} \mathcal{F}_{ii}
\end{aligned}
\tag{4.19}
$$

and

$$
\begin{aligned}
\forall i \neq j, \Gamma_{ii,jj} = \Gamma_{ij,ij} = \Gamma_{ij,ji} &= M(x_i^2 x_j^2) - P(x_i^2 x_j^2) - Q(x_i^2 x_j^2) \\
&= M(x_i^2 x_j^2) - \widetilde{M}(x_i^2 x_j^2) = \delta
\end{aligned}
\tag{4.20}
$$

and

$$
\begin{aligned}
\forall i \neq j, \Gamma_{ii,ij} = \Gamma_{ii,ji} &= M(x_i^3 x_j) - P(x_i^3 x_j) - Q(x_i^3 x_j) \\
&= M(x_i^3 x_j) - \widetilde{M}(x_i^3 x_j) = \frac{\mathcal{E}'_{st}}{k-2} - \frac{k}{k-2} \mathcal{F}_{st}
\end{aligned}
\tag{4.21}
$$

where the last equality uses equation (4.15).

Now we are ready to prove PSDness of $\Gamma$. We further decompose $\Gamma$ as $\Gamma = \Gamma' + \mathrm{blkdiag}(\Lambda', 0)$ where $\Lambda'$ is the $p \times p$ matrix with $\Lambda' = \delta \mathbf{1}\mathbf{1}^T$ [5]. Note that $\Lambda'$ is a PSD matrix and therefore it suffices to prove that $\Gamma' = \Gamma - \mathrm{blkdiag}(\Lambda', 0)$ is a PSD matrix.

Note that $\Gamma'$ has $ij$-th column the same as $ji$-th column, and therefore it's only of rank at most $p + p(p-1)/2$. We define $\Gamma''$ be the $p + p(p-1)/2$ by $p + p(p-1)/2$ submatrix of $\Gamma'$, that is indexed by subsets $(i,i)$ for $i \in [p]$ and $(i,j)$ for $i < j$. Therefore it suffices to prove that $\Gamma''$ is PSD. We prove it using Gershgorin Circle Theorem.

Note that by equation (4.19), we have that $\Gamma''_{ii,ii} = \Gamma'_{ii,ii} - \Lambda'_{ii,ii} = \frac{(p-k)}{k-1} \cdot \delta + \frac{1}{k-1} \mathcal{E}'_{ii} - \frac{k}{k-1} \mathcal{F}_{ii}$. Therefore by the lower bound for $\delta$ and Lemma 4.6, we obtain, $\Gamma''_{ii,ii} \geq (1-o(1))\frac{\epsilon k}{p}$. Moreover, $\Gamma''_{ii,ij} = \Gamma_{ii,ij} = \frac{\mathcal{E}'_{st}}{k-2} - \frac{k}{k-2} \mathcal{F}_{st}$ and therefore $|\Gamma''_{ii,ij}| \leq |\frac{\mathcal{E}'_{st}}{k-2}| + |\frac{k}{k-2} \mathcal{F}_{st}| \leq \widetilde{O}(\frac{\gamma}{n^{1.5}}) + \widetilde{O}\left( \frac{\gamma k^2 \sqrt{n}}{p^3} \right) \leq \widetilde{O}(\frac{\gamma}{n^{1.5}})$. Furthermore, for $i < j$, $\Gamma''_{ij,ij} = \Gamma_{ij,ij} = \delta \geq (1-o(1))\frac{\epsilon k^2}{p^2}$. Finally observe that $\Gamma''_{ii,jj} = 0$ by definition and all other entries of $\Gamma''$ are trivially 0 because the corresponding entries of $\Gamma$ and $\Lambda'$ vanish. Therefore we are ready to use a variant of Gershgorin Circle Theorem (Lemma 8.8) to prove the PSDness of $\Gamma'$. Taking $\alpha = 1/\gamma^2$, we have for any $i$,

$$\alpha \sum_{s,t:(s,t)\neq(i,i),s<t} |\Gamma''_{ii,st}| = \sum_{j\in[p]} |\Gamma''_{ii,jj}| + \sum_{j:j>i} |\Gamma''_{ii,ij}| + \sum_{j:j<i} |\Gamma''_{ii,ji}|$$

$$\leq \alpha p \cdot \widetilde{O}(\frac{\gamma}{n^{1.5}}) = o\left(\frac{k}{p}\right) \leq \Gamma''_{ii,ii}$$

where we used the fact that $k \gg \gamma\sqrt{n}$ and $\epsilon$ is a constant.

Moreover, for any $i < j$, we have that

$$\frac{1}{\alpha} \sum_{(s,t):(s,t)\neq(i,j),s<t} |\Gamma''_{ij,st}| \leq |\Gamma''_{ij,ii}| + |\Gamma''_{ij,jj}|$$

$$\leq \widetilde{O}(\frac{\gamma^3}{n^{1.5}}) = o\left(\frac{k^2}{p^2}\right) \leq \Gamma''_{ij,ij}$$

where we used $k \geq \gamma^4$ and $k \gg \gamma\sqrt{n}$. Therefore by Lemma 8.8, we obtain that $\Gamma''$ is PSD.

- Constraint (C2): Using Lemma 4.2 and equation (4.12), we have that

$$\sum_{s,t} |M(x_s x_t)| \leq \sum_{ss} M(x_s^2) + \sum_{s\neq t} |M(x_s x_t) - \widetilde{M}(x_s x_t)| + \sum_{s\neq t} |\widetilde{M}(x_s x_t)|$$

$$\leq k + p^2 \widetilde{O}(\frac{\gamma}{n^{1.5}}) + k^2/2 \leq k^2$$

- Constraint (C5): Finally, we check that $M$ satisfies the sparsity constraint (C5).

$$\sum_{i,j,s,t} |M(x_i x_j x_s x_t)| \leq \sum_{i,j,s,t} |\Gamma_{ij,st}| + \sum_{i,j,s,t} |\widetilde{M}(x_i x_j x_s x_t)|$$

$$\leq k^4$$

where we used (4.11) and the (trivial) facts that $\Gamma_{ij,st} \leq O(k/p)$ for any $i, j, s, t$ and there are only at most $O(p^2)$ nonzero entries in $\Gamma$.

- Objective value (Obj): Note that by constraint (C1) and Lemma 4.2, we have that $\sum_i M(x_i^2)\hat{\Sigma}_{ii} = k \geq \sum_i \widetilde{M}(x_i^2)\hat{\Sigma}_{ii}$, then

$$\sum_{i,j} M(x_i x_j)\hat{\Sigma}_{i,j} \geq \sum_{i,j} \widetilde{M}(x_i x_j)\hat{\Sigma}_{i,j} - \sum_{i\neq j} |M(x_i x_j) - \widetilde{M}(x_i x_j)||\hat{\Sigma}_{ij}|$$

$$\geq (1 - o(1))\gamma k - p^2 \cdot \widetilde{O}\left(\frac{\gamma}{n\sqrt{n}}\right) \cdot \widetilde{O}\left(\frac{1}{\sqrt{n}}\right)$$

$$\geq (1 - o(1))\gamma k$$

where in the second inequality we used Lemma 4.2 and the facts that $\hat{\Sigma}_{ij} = \frac{1}{n}\langle X_i, X_j \rangle \leq \widetilde{O}(1/\sqrt{n})$ and $|\mathcal{E}'|_{ij} + |\mathcal{F}'_{ij}| \leq \widetilde{O}\left(\frac{\gamma}{n^{1.5}}\right)$, and the last line uses the fact taht $\gamma^4 \leq k$.

$\square$

# 5 Proof of Theorem 3.1 and Theorem 3.2

In this section, we prove our main Theorems using the technical results of the previous sections. Before getting in the proof, we start with the observation that in order to get a lower bound of objective value $10\lambda k$, it suffices to consider the special case when $p = 10\lambda n$. The reason is that the objective value of $\text{SoS}_4$ is increasing in $p$ while fixing all other paramters: Suppose $p' \leq p$ and $\hat{\Sigma}' \in \mathbb{R}^{p' \times p'}$ is a submatrix of $\hat{\Sigma}$, and $M^{*'} : \mathbb{R}_{p'}[x]_4$ is the maximizer of $\text{SoS}_4(\hat{\Sigma}')$. Then we can extend $M'$ to $M : \mathbb{R}_p[x]_4 \to \mathbb{R}$ by simply defining that $M(x^S) = M^{*'}(x^S)$ if $S \subset [p']$ and 0 otherwise. This preserves all the constraint and objective value. Thus we proved that the objective value for $\Sigma$ is at least the one for $\Sigma'$. Formally, we have

**Proposition 5.1.** Fixing $\lambda, k, n$, given a data matrix $X \in \mathbb{R}^{p \times n}$, and any submatrix matrix $Y \in \mathbb{R}^{p' \times n}$ of $X$ with $p' \leq p_1$, let $\hat{\Sigma}_X$ and $\hat{\Sigma}_Y$ be the covariance matrices of $X$ and $Y$, then we have that $\text{SoS}_4(\hat{\Sigma}_X) \geq \text{SoS}_4(\hat{\Sigma}_Y)$.

Now we are ready to prove our main Theorem 3.1. The idea is very simple: we normalize the data matrix $X$ so that the resulting matrix $\bar{X}$ satisfies the the pseudorandomness condition 4.1. Then we apply Theorem 4.7 and obtain a moment matrix which give large objective value with respect to $\bar{X}$. Then we argue that the difference between $\bar{X}$ from $X$ is negligible so that the same moment matrix has also large objective value with respect to $X$.

*Proof of Theorem 3.1.* Using the observation above, we take $p = 1.1\gamma n$ with $\gamma = 11\lambda$, and we define $\bar{X}$ to matrix obtained by normalizing rows of $X$ to euclidean norm $\sqrt{n}$. Then by Theorem 7.1 it satisfies the pseudorandomenss condition 4.1. Let $\hat{\Sigma}' = \frac{1}{n}\bar{X}\bar{X}^T$ be the covariance matrix defined by $\bar{X}$. By Theorem 4.7 we have that $\text{SoS}_4(\hat{\Sigma}') \geq (1 - o(1))\gamma k \geq 11\lambda k$. Moreover, let $M$ be the moment defined in Theorem 4.7, and $M_2$ its restriction to degree-2 moments, that is, $M_2 = (\text{mat}(M)_{S,T})_{|S|=|T|=1}$. We are going to show that the entry-wise difference between $\hat{\Sigma}$ and $\hat{\Sigma}'$ are small enough so that $\langle M_2, \hat{\Sigma} \rangle$ is close to $\langle M_2, \hat{\Sigma}' \rangle$.

Note that since $\|X_i\|^2 = n \pm \widetilde{O}(\sqrt{n})$, therefore for any $i \neq j$, $\hat{\Sigma}'_{ij} = \frac{\hat{\Sigma}_{ij}}{\|X_i\|\|X_j\|} = \hat{\Sigma}_{ij} \pm \widetilde{O}(\frac{1}{\sqrt{n}})|\hat{\Sigma}|_{ij} = \hat{\Sigma}_{ij} \pm \widetilde{O}(\frac{1}{n})$. For $i = j$, we have similarly that $\hat{\Sigma}'_{ii} = \hat{\Sigma}_{ii} \pm \widetilde{O}(\frac{1}{\sqrt{n}})$. We bound the difference between $\langle M_2, \hat{\Sigma}' \rangle$ and $\langle M_2, \hat{\Sigma}' \rangle$ by the sum of the entry-wise differences:

$$|\langle M_2, \hat{\Sigma}' - \hat{\Sigma} \rangle| \leq \sum_i M(x_i^2)|\hat{\Sigma}_{ii} - \hat{\Sigma}'_{ii}| + \sum_{i \neq j} M(x_i x_j)|\hat{\Sigma}_{ij} - \hat{\Sigma}'_{ij}|$$

$$\leq p \cdot O(k/p) \cdot \widetilde{O}(\frac{1}{\sqrt{n}}) + p^2 \cdot \widetilde{O}(\frac{\gamma k \sqrt{n}}{p^2}) \cdot \widetilde{O}(\frac{1}{n}) = o(k)$$

Therefore $\langle M_2, \hat{\Sigma} \rangle \geq (1-o(1))\gamma k - o(k) = (1-o(1))\gamma k$. Therefore the moment $M$ gives objective value $(1 - o(1))\gamma k$ for data $\hat{\Sigma}$, and therefore $\text{SoS}_4(\hat{\Sigma}) \geq (1-o(1))\gamma k \geq 10\lambda k$. □

Then we prove that Theorem 3.1 together with the arguments in [KNV15] implies Theorem 3.2. The intuition behind is the following: Suppose $M_2^*$ is very close to $vv^T$, then it is close to rank-1 and its leading eigenvector is close to $\hat{v}$. However, since we prove that the objective value is large (which is true also in the planted vector case), $M_2^*$ needs to be highly correlated with $\hat{\Sigma}$, which implies its leading eigenvector $\hat{v}$ needs to be correlated with $\hat{\Sigma}$, which in turns implies that $v$ is correlated with $\hat{\Sigma}$. However, it turns out that $v$ is not correlated enough with $\hat{\Sigma}$, which leads to a contradiction.

*Proof of Theorem 3.2.* We first prove that the optimal value of $\text{SoS}_4(\hat{\Sigma})$ for hypothesis $H_v$ is also at least $0.99kp/n$. Suppose $v$ has support $S$ of size $k$. We consider the restriction of linear operator $M$ to the subset $T = [p]\backslash S$, denoted by $M_T$. That is, we have that $M_T(x^\alpha) = 0$ for any monomial $x^\alpha$ that contains a factor $x_i$ with $i \in S$, and otherwise $M_T(x^\alpha) = M(x^\alpha)$. We also consider the data matrix $X_T$ obtained by restricting to the rows indexed by $T$. Note that since $X_T$ doesn't contain the signal, and $k \gg \sqrt{n}$), using Theorem 4.7 with $\gamma = (p - k)/(1.01n)$, we have that there exists pseudo-moment $M_T^*$ which gives objective value $\geq (1-o(1))\gamma k \geq 0.99pk/n$ with respective to covariance matrix $\hat{\Sigma}_T = \frac{1}{n}X_T X_T^T$. Note that by Proposition 5.1, $\text{SoS}_4(\hat{\Sigma}) \geq \text{SoS}_4(\hat{\Sigma}_T)$ and therefore we obtain that under hypothesis $H_v$, with high probability, $\text{SoS}_4(\hat{\Sigma}) \geq 0.99kp/n$.

Now suppose $M^*$ is the maximizer of $\text{SoS}_4(\hat{\Sigma})$, and $M_2^* = (M^*(x_i x_j))_{i,j \in [p]}$. For the sake of contradiction, we assume that $\|\frac{1}{k}M_2^* - vv^T\| \leq 1/5$. We first show that this implies that $M_2^*$ has an eigenvector $\hat{v}$ that is close to $v$ and its eigenvalue is large. Indeed we have $\|\frac{1}{k}M_2\| \geq \|vv^T\| - 1/5 = 4/5$. Therefore the top eigenvector of $\frac{1}{k}M_2^*$ has eigenvalue larger than $4/5$. Then we can decompose the difference between $\frac{1}{k} \cdot M_2^*$ and $vv^T$ into $\frac{1}{k} \cdot M_2^* - vv^T = \frac{4}{5} \cdot (\hat{v}\hat{v}^T - vv^T) + ((\frac{1}{k}M_2^* - \frac{4}{5}\hat{v}\hat{v}^T) - \frac{1}{5}vv^T)$. Note that since $(\frac{1}{k}M_2^* - \frac{4}{5}\hat{v}\hat{v}^T)$ is a PSD matrix with eigenvalue at most $1/5$, we have $\|((\frac{1}{k}M_2^* - \frac{4}{5}\hat{v}\hat{v}^T) - \frac{1}{5}vv^T)\| \leq 2/5$ by triangle inequality. Then by triangle inequality and our assumption again we obtain that

$$\frac{1}{5} \geq \|\frac{1}{k} \cdot M_2^* - vv^T\| \geq \|\frac{4}{5}(\hat{v}\hat{v}^T - vv^T)\| - \|\left(\left(\frac{1}{k}M_2^* - \frac{4}{5}\hat{v}\hat{v}^T\right) - \frac{1}{4}vv^T\right)\| \geq \|\frac{4}{5}(\hat{v}\hat{v}^T - vv^T)\| - \frac{2}{5}$$

Therefore we obtain that $\|vv^T - \hat{v}\hat{v}^T\| \leq 3/5$ and therefore $\|vv^T - \hat{v}\hat{v}^T\|_F^2 \leq 2\|vv^T - \hat{v}\hat{v}^T\|^2 \leq 1$. It follow that $|\langle v, \hat{v}\rangle|^2 = 1 - \frac{1}{2}\|vv^T - \hat{v}\hat{v}^T\|_F^2 \geq 1/2$.

Next we are going to show that it is impossible for $M_2^*$ to have an eigenvector that is close to $v$ with a large eigenvalue and therefore we will get a contradiction. Let $\hat{v} = \alpha v + \beta s$ where $s$ is orthogonal to $v$ and $\alpha^2 + \beta^2 = 1$ and $\alpha \geq \sqrt{1/2}$, and $\beta \leq \sqrt{1/2}$. Then using triangle inequality we have that $\|\hat{v}\|_{\hat{\Sigma}} \leq \|\alpha v\|_{\hat{\Sigma}} + \|\beta s\|_{\hat{\Sigma}} \leq \sqrt{O(\lambda)}\alpha + \sqrt{\|\hat{\Sigma}\|}\beta$. Proposition 5.3 of [KNV15] implies that for sufficiently large $C$ and $\lambda \geq 1$, when $p/n \geq C\lambda$, $\|\hat{\Sigma}\| \leq 1.01p/n$. Therefore under our assumption we have that $\|\hat{\Sigma}\| \leq 1.01p/n$. It follows $\beta \leq \sqrt{1/2}$ that $\|\hat{v}\|_{\hat{\Sigma}} \leq \sqrt{O(\lambda)}\alpha + \sqrt{\beta} \cdot \sqrt{p/n} \leq \sqrt{O(\lambda)} + \sqrt{p/2n}$. Therefore, we have that

$$\begin{aligned}
0.99p/n \leq \frac{1}{k} \cdot \text{SoS}_4(\hat{\Sigma}) = \frac{1}{k} \cdot \langle M_2, \hat{\Sigma}\rangle &= \frac{1}{k} \cdot \langle M_2^* - \frac{4k}{5} \cdot \hat{v}\hat{v}^T, \hat{\Sigma}\rangle + \frac{4}{5}\langle \hat{v}\hat{v}^T, \hat{\Sigma}\rangle \\
&\leq \frac{1}{k}\text{tr}(M_2^* - \frac{4}{5} \cdot \hat{v}\hat{v}^T)\|\hat{\Sigma}\|_2 + \frac{4}{5}\|\hat{v}\|_{\hat{\Sigma}}^2 \\
&\leq \frac{1}{5}\|\hat{\Sigma}\| + O(\alpha^2\lambda) + O(\sqrt{\lambda p/n}) + \frac{2}{5} \cdot p/n \\
&\leq \frac{4}{5} \cdot \frac{p}{n} + O(\sqrt{\lambda p/n})
\end{aligned}$$

where in the third line we used the fact that $\|\hat{v}\|_{\hat{\Sigma}} \leq \sqrt{O(\lambda)} + \sqrt{p/2n}$, and the last line we used $\|\hat{\Sigma}\| \leq 1.01p/n$. Note that this is a contradiction since we assumed that $p/n \geq C\lambda$ for sufficiently large $C$. $\qquad\square$

# 6  Analysis of matrices $P$ and $Q$

In this section we prove Lemma 4.4 and 4.5. They basically follow direct calculation and the pseudorandomness properties of data matrix $X$ listed in Condition 4.1.

*Proof of Lemma 4.4.* Note that since $p = 1.1\gamma n$ and $1 \leq \gamma \leq n$, we have that $O(n^2) \geq p \geq n$. We verify equations (4.3), (4.4) and (4.5) and the bounds for $\mathcal{F}$ and $\mathcal{E}$ one by one.

- Equation (4.3):

  For the case when $i = j$, we verify $P(x_i^4)$ using property (P1) and (P2),

  $$
  \begin{aligned}
  P(x_i^4) &= \frac{\gamma k}{p^2 n^3}\left(\langle X_i, X_i\rangle^4 + \sum_{\ell \in [p] \setminus i}\langle X_i, X_\ell\rangle^4\right) \\
  &\leq \frac{\gamma k}{p^2 n^3}\left(n^3\langle X_i, X_i\rangle + \widetilde{O}(pn^2)\right) \\
  &= \widetilde{M}(x_i^2) + \widetilde{O}\left(\frac{\gamma k}{pn}\right)
  \end{aligned}
  $$

  For distinct $i, j$, we have that

  $$
  \begin{aligned}
  P(x_i^3 x_j) &= \frac{\gamma k}{p^2 n^3}\left(\langle X_i, X_i\rangle^3\langle X_i, X_j\rangle + \langle X_i, X_j\rangle^3\langle X_i, X_i\rangle + \sum_{\ell \in [p]\setminus i \cup j}\langle X_i, X_\ell\rangle^3\langle X_j, X_\ell\rangle\right) \\
  &= \frac{\gamma k}{p^2 n^3}\left(n^3\langle X_i, X_j\rangle \pm \widetilde{O}(n^{2.5}) \pm \widetilde{O}(pn^{1.5})\right) \\
  &= \widetilde{M}(x_i x_j) \pm \widetilde{O}\left(\frac{\gamma k}{p^2 n^{.5}}\right) \pm \widetilde{O}\left(\frac{\gamma k}{pn^{1.5}}\right) \\
  &= \widetilde{M}(x_i x_j) \pm \widetilde{O}\left(\frac{\gamma k}{pn^{1.5}}\right)
  \end{aligned}
  $$

  where in the second equality we use equation (P3), and $p \geq n$.

- Equation (4.5):

  Note that for distinct $i, j, s, t$, by equation (P4), we have

  $$
  |P(x_i x_j x_s x_t)| = \frac{\gamma k}{p^2 n^3}\left|\sum_{\ell \in [p]}\langle X_i, X_\ell\rangle\langle X_j, X_\ell\rangle\langle X_s, X_\ell\rangle\langle X_t, X_\ell\rangle\right| \leq \widetilde{O}\left(\frac{\gamma k}{p^{1.5}n}\right)
  $$

  Therefore taking the sum over all distinct $i, j, s, t$ we have

  $$
  \sum_{i,j,s,t \text{ distinct}} |P(x_i x_j x_s x_t)| \leq \widetilde{O}\left(\frac{\gamma k}{p^{1.5}n}\right)\cdot p^4 = \widetilde{O}\left(\frac{\gamma p^{2.5} k}{n}\right) \leq k^4/4 \tag{6.1}
  $$

  where we used $k \gg \gamma^{7/6}\sqrt{n}$, which implies that $k^3 \gg \gamma p^{2.5}/n$.

By equation (4.2) and equation (4.3), we have that

$$\sum_{i,j} |P(x_i^3 x_j)| \leq \sum_{i,j} |\widetilde{M}(x_i x_j)| + p^2 \cdot \widetilde{O}\left(\frac{\gamma k}{pn^{1.5}}\right) + p \cdot \widetilde{O}\left(\frac{\gamma k}{pn}\right) \leq k^2/2 + \widetilde{O}\left(\gamma k \sqrt{n}\right) \leq k^2 \quad (6.2)$$

where we used the fact that $p \leq n^2$ and $k \gg \gamma \sqrt{n}$.

For distinct $i, s, t$, we have that

$$\left| \sum_{\ell \in [p]} \langle X_i, X_\ell \rangle^2 \langle X_s, X_\ell \rangle \langle X_t, X_\ell \rangle \right| = \left| \sum_{\ell \neq i,s,t} \langle X_i, X_\ell \rangle^2 \langle X_s, X_\ell \rangle \langle X_t, X_\ell \rangle \right| + \left| \langle X_i, X_i \rangle^2 \langle X_s, X_i \rangle \langle X_t, X_i \rangle \right|$$
$$+ \left| \langle X_i, X_s \rangle^2 \langle X_s, X_s \rangle \langle X_t, X_s \rangle \right| + \left| \langle X_i, X_t \rangle^2 \langle X_s, X_t \rangle \langle X_t, X_t \rangle \right|$$
$$= \widetilde{O}(pn^2) + \widetilde{O}(n^3) + \widetilde{O}(n^{2.5}) = \widetilde{O}(pn^2)$$

It follows that

$$|P(x_i^2 x_s x_t)| = \frac{\gamma k}{p^2 n^3} \left| \sum_{\ell \in [p]} \langle X_i, X_\ell \rangle^2 \langle X_s, X_\ell \rangle \langle X_t, X_\ell \rangle \right| \leq \widetilde{O}\left(\frac{\gamma k}{pn}\right)$$

and therefore,

$$\sum_{i,s,t \text{ disctinct}} |P(x_i^2 x_s x_t)| \leq p^3 \cdot \widetilde{O}\left(\frac{\gamma k}{pn}\right) = \widetilde{O}\left(\frac{\gamma k p^2}{n}\right) \quad (6.3)$$

Therefore, combining equation (6.1), (6.2), (6.3), we obtain that

$$\sum_{i,j,s,t} |P(x_i x_j x_s x_t)| \leq k^2 + \widetilde{O}\left(\frac{\gamma k p^2}{n}\right) + k^4/4 \leq k^4/3$$

- Equation (4.4):

  Finally it remains to bound $\mathcal{E}$. Note that $\mathcal{E}$ is a sum of submatrices of $P$ and therefore it is PSD. Moreover,

$$\mathcal{E}_{ss} = \sum_{i \in [p]} P(x_i^2 x_s^2) = \frac{\gamma k}{p^2 n^3} \sum_i \sum_\ell \langle X_i, X_\ell \rangle^2 \langle X_s, X_\ell \rangle^2$$
$$= \frac{\gamma k}{p^2 n^3} \sum_i \langle X_i, X_\ell \rangle^2 \sum_\ell \langle X_s, X_\ell \rangle^2$$
$$\leq \frac{\gamma k}{p^2 n^3} \cdot \widetilde{O}(p^2 n^2) = \widetilde{O}\left(\frac{\gamma k}{n}\right)$$

where the last inequality uses equation (P2). Finally we bound $\mathcal{E}_{st}$ using equation (P6)

$$
\begin{aligned}
\sum_{i \in [p]} P(x_i^2 x_s x_t) &= \frac{\gamma k}{p^2 n^3} \sum_i \sum_\ell \langle X_i, X_\ell \rangle \langle X_i, X_\ell \rangle \langle X_s, X_\ell \rangle \langle X_t, X_\ell \rangle \\
&\leq \frac{\gamma k}{p^2 n^3} \widetilde{O}(p^2 n^{1.5}) = \widetilde{O}(\frac{\gamma k}{n^{1.5}})
\end{aligned}
$$

$\square$

*Proof of Lemma 4.5.* Again we verify equation (4.6), (4.7) and (4.8) in order.

- Equation (4.6): By definition we have that for any $i, j$,

$$
Q(x_i^3 x_j) = \frac{\gamma k^2}{p^3 n} \cdot 3n \langle X_i, X_j \rangle = \frac{3\gamma k^2}{p^3} \langle X_i, X_j \rangle = \frac{3k}{p} \widetilde{M}(x_i x_j)
$$

- Equation (4.8): For the sparsity constraint, we note first that for distinct $i, j, s, t$, using property (P2), we have

$$
|Q(x_i x_j x_s x_t)| \leq \frac{\gamma k^2}{p^3 n} \cdot \widetilde{O}(n) = \widetilde{O}\left( \frac{\gamma k^2}{p^3} \right)
$$

and therefore taking sum, we have

$$
\sum_{i,j,s,t \text{ disctinct}} |Q(x_i x_j x_s x_t)| \leq \widetilde{O}(\gamma k^2 p) \leq k^4/6 \tag{6.4}
$$

where we used the fact that $k^2 \gg = \gg c^2 n$. We bound other terms as follows:

For any $i, j, s, t \in [p]$, we have that

$$
Q(x_i x_j x_s x_t) \leq \frac{\gamma k^2}{p^3 n} \cdot 3n^2 = \frac{3\gamma k^2 n}{p^3}
$$

There are only at most $O(p^3)$ different choices of $i, j, s, t$ such that $i, j, s, t$ are not distinct, therefore we have

$$
\sum_{i,j \text{ not distinct}} |Q(x_i^3 x_j)| \leq \frac{3\gamma k^2 n}{p^3} \cdot O(p^3) \leq k^4/6 \tag{6.5}
$$

where we used the fact that $k \gg \gamma \sqrt{n}$ and $\gamma \geq 1$.

Combining equation (6.4) and (6.5), we obtain that

$$
\sum_{i,j,s,t} |Q(x_i x_j x_s x_t)| \leq k^4/3
$$

- Equation (4.7): For any $s, t$, we have

$$
\begin{aligned}
\sum_i Q(x_i^2 x_s x_t) &= \frac{\gamma k^2}{p^3 n} \sum_{i \in [p]} \left( n \langle X_s, X_t \rangle + 2 \langle X_i, X_s \rangle \langle X_i, X_t \rangle \right) \\
&= \frac{\gamma k^2}{p^2} \langle X_s, X_t \rangle + \frac{2 \gamma k^2}{p^3 n} \sum_{i \in [p]} \langle X_i, X_s \rangle \langle X_i, X_t \rangle
\end{aligned}
$$

Therefore $\mathcal{G}_{st} = \frac{2\gamma k^2}{p^3 n} \sum_{i \in [p]} \langle X_i, X_s \rangle \langle X_i, X_t \rangle$ forms a PSD matrix. Moreover, when $s \neq t$, using equation (P5), we have that

$$
\begin{aligned}
\sum_i Q(x_i^2 x_s x_t) &= k \widetilde{M}(x_s x_t) \pm \frac{2\gamma k^2}{p^3 n} \cdot \widetilde{O}(p\sqrt{n}) \\
&= k \widetilde{M}(x_s x_t) \pm \widetilde{O}\left( \frac{\gamma k^2}{p^2 \sqrt{n}} \right)
\end{aligned}
$$

When $s = t$, we have that

$$
\begin{aligned}
\sum_i Q(x_i^2 x_s^2) &= k \widetilde{M}(x_s x_t) \pm \frac{2\gamma k^2}{p^3 n} \cdot \widetilde{O}(pn) \\
&= k \widetilde{M}(x_s x_t) \pm \widetilde{O}\left( \frac{\gamma k^2}{p^2} \right)
\end{aligned}
$$

$\square$

# 7 Pseudo-randomness of $X$

In this section, we prove that basically as long as the noise model is subgaussian and has variance 1(which generalizes the standard Bernoulli and Gaussian distributions), after normalizing the rows of the data matrix $X \sim H_0$, it satisfies the pseudorandomness condition 4.1.

**Theorem 7.1.** Suppose independent random variables $X_1, \ldots, X_p \in \mathbb{R}^n$ satisfy that for any $i$, $X_i$ has a i.i.d entries with mean zero, variance 1, and subgaussian variance proxy[6] $O(1)$, then the matrix $\bar{X}$ with $\frac{X_i^T}{\|X_i\|}$ as rows satisfies the pseudorandomness condition 4.1.

The proof of the Theorem relies on the following Proposition and Theorem 7.4. The proposition says that $\frac{X_i^T}{\|X_i\|}$ still satisfies *good* properties like symmetry and that each entries has a subgaussian tail, even though its entries are no longer independent due to normalization. These properties will be encapsulated in the definition of a *good* random variable following the proposition. Then we prove in Theorem 7.4 that these properties suffice for establishing the pseudorandomness Condition 4.1 with high probability. We will heavily use the $\psi_\alpha$-Orlicz norm (denoted $\|\cdot\|_{\psi_\alpha}$) of a random variable, defined in Definition 8.1, and its properties, summarized in the next (toolbox) section. Intuitively, $\|\cdot\|_{\psi_2}$ norm is a succinct and convenient way to capture the "subgaussianity" of a random variable.

**Proposition 7.2.** Suppose $y \in \mathbb{R}^n$ has i.i.d entries with variance 1 and mean zero, and gaussian variance proxy $O(1)$, then random variable $x = \frac{y}{\|y\|}$ satisfies the following properties:

1. $\|x\|^2 = n$, almost surely.

2. for any vector $a \in \mathbb{R}^n$ with $\|a\|^2 \leq 2n$, $\|\langle x, a \rangle\|_{\psi_2}^2 \leq O(n)$.

3. $\|\,|x|_\infty\|_{\psi_2} \leq \widetilde{O}(1)$

4. $\mathbb{E}[x_i^2] = 1$, $\mathbb{E}[x_i^4] = C_4$, and $\mathbb{E}[x_i^2 x_j^2] = C_{2,2}$ for all $i$ and $j \neq i$, where $C_4, C_{2,2} = O(1)$ are constants that don't depend on $i, j$

5. For any monomial $x^\alpha$ with an odd degree, $\mathbb{E}[x^\alpha] = 0$.

For simplicity, we call a random variable *good* if it satisfies the five properties listed in the proposition above. Goodness will be invoked in most statements below.

**Definition 7.3** (goodness). A random variable $x \in \mathbb{R}^n$ is called *good*, if it satisfies the conclusion of Proposition 7.2.

We will show a random matrix $X$ with *good* rows satisfies the pseudo-randomness Condition 4.1 with high probability.

**Theorem 7.4.** Suppose independent $n$-dimensional random vectors $X_1, \ldots, X_p$ with $p \geq n$ are all *good*, then $X_1, \ldots, X_p$ satisfies the pseudorandomness condition 4.1 with high probability.

The general approach to prove the theorem is just to use the concentration of measure. The only caveat here is that in most of cases, the random variables that we are dealing with are not bounded a.s. so we can't use Chernoff bound or Bernstein inequality directly. However, though these random variables are not bounded a.s., they typically have a light tail, that is, their $\psi_\alpha$ norms can be bounded. Then we are going to apply Theorem 8.4 of Ledoux and Talagrand's, a extended version of Bernstein inequality with only $\psi_\alpha$ norm boundedness required. We will also use other known technical results listed in the toolbox Section 8.

*Proof of Theorem 7.4.* Equation (P1) and (P2) follows the assumptions on $X_i$'s and union bound. Equation (P3) is proved in Lemma 7.5 by taking $u = X_s$ and $v = X_t$ and view the rest of $X_i$;s as $Z_j$'s in the statement of Lemma 7.5. Equation (P4) is proved in Lemma 7.6, (P5) in Lemma 7.8, (P6) in Lemma 7.10, and equation P7 is proved in Lemma 7.15. □

**Lemma 7.5.** For any *good* random variable $x$, we have that for fixed $u, v$ with $\|u\|^2 = \|v\|^2 = n$, $|u|_\infty \leq \widetilde{O}(1)$, $|v|_\infty \leq \widetilde{O}(1)$, and $\langle u, v \rangle \leq \widetilde{O}(\sqrt{n})$,

$$\left| \mathbb{E}\left[ \langle x, u \rangle^3 \langle x, v \rangle \right] \right| \leq \widetilde{O}(n^{1.5})$$

and moreover, for $p \geq n$ and a sequence of *good* independent random variables $Z_1, \ldots, Z_p$, we have that with high probability,

$$\left| \sum_{i=1}^p \langle Z_i, u \rangle^3 \langle Z_i, v \rangle \right| \leq \widetilde{O}(n^{1.5} p)$$

*Proof.* We calculate the expectation as follows

$$
\mathbb{E}\left[\langle x, u\rangle^3 \langle x, v\rangle\right] = \mathbb{E}\left[\left(\sum_i u_i^2 x_i^2 + 2\sum_{i<j} u_i u_j x_i x_j\right)\left(\sum_i v_i u_i x_i^2 + \sum_{i\neq j} u_i v_j x_i x_j\right)\right]
$$

$$
= \mathbb{E}\left[\sum_i u_i^3 v_i x_i^4\right] + \mathbb{E}\left[\sum_{i\neq j} u_i^2 u_j v_j x_i^2 x_j^2\right] + \mathbb{E}\left[\sum_{i\neq j} u_i u_j (u_i v_j + v_i u_j) x_i^2 x_j^2\right]
$$

$$
= (C_4 - C_{2,2})\sum_i u_i^3 v_i + C_{2,2} n \sum_i u_i v_i + C_{2,2}\sum_{i\neq j}(u_i^2 u_j v_j + u_j^2 u_i v_i)
$$

$$
= (C_4 - 3C_{2,2})\sum_i u_i^3 v_i + 3C_{2,2} n \sum_i u_i v_i
$$

Therefore by our assumption on $u$ and $v$ we obtain that

$$
\left|\mathbb{E}\left[\langle x, u\rangle^3 \langle x, v\rangle\right]\right| \leq \widetilde{O}(n) + O(n)|\langle u, v\rangle| \leq \widetilde{O}(n^{1.5})
$$

Now we prove the second statement. Since $\|\langle Z_i, u\rangle\|_{\psi_2} \leq O(\sqrt{n})$, by Lemma 8.5 we have that $\|\langle Z_i, u\rangle^3 \langle Z_i, v\rangle\|_{\psi_{1/2}} \leq O(n^2)$, and it follows Lemma 8.6 that $\|\langle Z_i, u\rangle^3 \langle Z_i, v\rangle - \mathbb{E}\left[\langle Z_i, u\rangle^3 \langle Z_i, v\rangle\right]\|_{\psi_{1/2}} \leq O(n^2)$ Then by Lemma 8.4 we obtain that with high probability,

$$
\sum_{i=1}^p \langle Z_i, u\rangle^3 \langle Z_i, v\rangle - \mathbb{E}\left[\sum_{i=1}^p \langle Z_i, u\rangle^3 \langle Z_i, v\rangle\right] \leq \widetilde{O}(n^2 \sqrt{p})
$$

Note that we have proved that $\left|\mathbb{E}\left[\sum_{i=1}^p \langle Z_i, u\rangle^3 \langle Z_i, v\rangle\right]\right| = \widetilde{O}(n^{1.5})$, therefore we obtain the desired result. $\square$

**Lemma 7.6.** Suppose $p \geq n$ and $X_1, \ldots, X_p$ are *good* independent random variables, then with high probability, for any distinct $i, j, s, t$,

$$
\left|\sum_{\ell\in[p]} \langle X_i, X_\ell\rangle \langle X_j, X_\ell\rangle \langle X_s, X_\ell\rangle \langle X_t, X_\ell\rangle\right| \leq \widetilde{O}(n^2 \sqrt{p})
$$

*Proof.* Fixing $i, j, s, t$, we can write

$$
\sum_{\ell\in[p]} \langle X_i, X_\ell\rangle \langle X_j, X_\ell\rangle \langle X_s, X_\ell\rangle \langle X_t, X_\ell\rangle = \sum_{\ell\in[p]\setminus\{i,j,s,t\}} \langle X_i, X_\ell\rangle \langle X_j, X_\ell\rangle \langle X_s, X_\ell\rangle \langle X_t, X_\ell\rangle
$$
$$
+ n\langle X_j, X_i\rangle \langle X_s, X_i\rangle \langle X_t, X_i\rangle + n\langle X_i, X_j\rangle \langle X_s, X_j\rangle \langle X_t, X_j\rangle
$$
$$
+ n\langle X_i, X_s\rangle \langle X_j, X_s\rangle \langle X_t, X_s\rangle + n\langle X_i, X_t\rangle \langle X_j, X_t\rangle \langle X_s, X_t\rangle
$$

Using Lemma 7.7, the first term on RHS is bounded by $\widetilde{O}(n^2 \sqrt{p})$ with high probability over the randomness of $X_\ell, \ell \in [p]\setminus\{i,j,s,t\}$. The rest of the four terms are bounded by $\widetilde{O}(n^{2.5})$. Therefore putting together $\|\sum_{\ell\in[p]} \langle X_i, X_\ell\rangle \langle X_j, X_\ell\rangle \langle X_s, X_\ell\rangle \langle X_t, X_\ell\rangle\| \leq \widetilde{O}(n^2 \sqrt{p})$ for any fixed $i, j, s, t$ with high probability and taking union bound we get the result. $\square$

**Lemma 7.7.** For any good random variable $x$, and for fixed $a, b, c, d$ such that $\max\{|a|_\infty, |b|_\infty, |c|_\infty, |d|_\infty\} = \widetilde{O}(1)$, and all the pair-wise inner products between $a, b, c, d$ have magnitude at most $\widetilde{O}(\sqrt{n})$, we have that

$$|\mathbb{E}\left[\langle x, a\rangle\langle x, b\rangle\langle x, c\rangle\langle x, d\rangle\right]| = \widetilde{O}(n)$$

and moreover, for $p \geq n$ and a sequence independent random variable $Z_1, \ldots, Z_p$ such that each $Z_i$ satisfies the conclusion of proposition 7.2, we have that with high probability,

$$\left|\sum_{i=1}^{p}\langle Z_i, a\rangle\langle Z_i, v\rangle\langle Z_i, c\rangle\langle Z_i, d\rangle\right| \leq \widetilde{O}(n^2\sqrt{p})$$

*Proof.* We calculate the mean

$$\mathbb{E}\left[\langle x, a\rangle\langle x, b\rangle\langle x, c\rangle\langle x, d\rangle\right] = \mathbb{E}\left[\sum_{i\in[p]} a_i b_i c_i d_i x_i^4 + \left\{\sum_{i\neq j} a_i b_i c_j d_j x_i^2 x_j^2\right\}\right]$$

where we use $\left\{\sum_{i\neq j} a_i b_i c_j d_j x_i^2 x_j^2\right\}$ to denote the sum of $a_i b_i c_j d_j x_i^2 x_j^2$ and all its permutations with repect to $a, b, c, d$.

Note that

$$\left|\mathbb{E}\left[\sum_{i\neq j} a_i b_i c_j d_j x_i^2 x_j^2\right]\right| = C_{2,2}\left|\langle a, b\rangle\langle c, d\rangle - \sum_{i\in[p]} a_i b_i c_i d_i\right| \leq \widetilde{O}(n)$$

and

$$\left|\mathbb{E}\left[\sum_{i\in[p]} a_i b_i c_i d_i x_i^4\right]\right| = \left|C_4 \sum_{i\in[p]} a_i b_i c_i d_i\right| \leq \widetilde{O}(n)$$

and therefore we have $|\mathbb{E}\left[\langle x, a\rangle\langle x, b\rangle\langle x, c\rangle\langle x, d\rangle\right]| \leq \widetilde{O}(n)$.

Since $\langle x, a\rangle$ has $\psi_2$ norm $\sqrt{n}$ and similar for the other three terms, we have that by Lemma 8.5 that $\|\langle x, a\rangle\langle x, b\rangle\langle x, c\rangle\langle x, d\rangle\|_{\psi_{1/2}} \leq O(n^2)$. Therefore using Theorem 8.4 we have that

$$\left\|\sum_{i=1}^{p}\langle Z_i, a\rangle\langle Z_i, v\rangle\langle Z_i, c\rangle\langle Z_i, d\rangle - \mathbb{E}[\sum_{i=1}^{p}\langle Z_i, a\rangle\langle Z_i, v\rangle\langle Z_i, c\rangle\langle Z_i, d\rangle]\right\|_{\psi_{1/2}} \leq \widetilde{O}(n^2\sqrt{p})$$

$\square$

**Lemma 7.8.** Suppose $p \geq n$ and $X_1, \ldots, X_p$ are *good* independent random variables, then with high probability, for any distinct $s, t$,

$$\left|\sum_{i\in[p]}\langle X_i, X_s\rangle\langle X_i, X_t\rangle\right| \leq \widetilde{O}(p\sqrt{n})$$

*Proof.* With high probability over the randomness of $X_i, i \in [p]\backslash\{s,t\}$,

$$\sum_{i\in[p]}\langle X_i, X_s\rangle\langle X_i, X_t\rangle = \sum_{i\in[p]\backslash\{s,t\}}\langle X_i, X_s\rangle\langle X_i, X_t\rangle + 2\langle X_s, X_t\rangle \leq \widetilde{O}(p\sqrt{n}) + \widetilde{O}(\sqrt{n})$$

where the last inequality is by Lemma 7.9. Taking union bound we complete the proof. $\square$

**Lemma 7.9.** For $p \geq n$ and a sequence of *good* independent random variable $Z_1, \ldots, Z_p$, and any two fixed vectors $u, v$ with $|u|_\infty \leq \widetilde{O}(1)$ and $|v|_\infty \leq \widetilde{O}(1)$, and $\langle u, v \rangle \leq \widetilde{O}(\sqrt{n})$, we have that with high probability,

$$\left| \sum_{i \in [p]} \langle Z_i, u \rangle \langle Z_i, v \rangle \right| \leq \widetilde{O}(p\sqrt{n})$$

*Proof.* $\mathbb{E}[\langle Z_i, u \rangle \langle Z_i, v \rangle] = \langle u, v \rangle \leq \widetilde{O}(\sqrt{n})$, and therefore $\left| \mathbb{E}\left[ \sum_{i \in [p]} \langle Z_i, u \rangle \langle Z_i, v \rangle \right] \right| \leq \widetilde{O}(p\sqrt{n})$. Note that $\|\langle Z_i, u \rangle\|_{\psi_2} \leq O(\sqrt{n})$ and therefore $\|\langle Z_i, u \rangle \langle Z_i, v \rangle\|_{\psi_1} \leq O(n)$. By Theorem 8.4, we have the desired result. $\qquad \square$

**Lemma 7.10.** Suppose $p \geq n$ and $X_1, \ldots, X_p$ are *good* independent random variables, then with high probability, for any distinct $s, t$,

$$\sum_{i, \ell \in [p]} \langle X_i, X_\ell \rangle \langle X_i, X_\ell \rangle \langle X_s, X_\ell \rangle \langle X_t, X_\ell \rangle \leq \widetilde{O}(p^2 n^{1.5})$$

*Proof.* We expand the target as follows:

$$\sum_{i, \ell \in [p]} \langle X_i, X_\ell \rangle \langle X_i, X_\ell \rangle \langle X_s, X_\ell \rangle \langle X_t, X_\ell \rangle = \sum_{i \in [p], \ell \in [p] \backslash s \cup t} \langle X_i, X_\ell \rangle \langle X_i, X_\ell \rangle \langle X_s, X_\ell \rangle \langle X_t, X_\ell \rangle$$

$$+ \sum_i \langle X_i, X_s \rangle^2 \langle X_s, X_s \rangle \langle X_t, X_s \rangle + \sum_i \langle X_i, X_t \rangle^2 \langle X_t, X_t \rangle \langle X_s, X_t \rangle$$

$$= \sum_{i \in [p] \backslash s \cup t, \ell \in [p] \backslash s \cup t} \langle X_i, X_\ell \rangle \langle X_i, X_\ell \rangle \langle X_s, X_\ell \rangle \langle X_t, X_\ell \rangle$$

$$+ \sum_i \langle X_i, X_s \rangle^2 \langle X_s, X_s \rangle \langle X_t, X_s \rangle + \sum_i \langle X_i, X_t \rangle^2 \langle X_t, X_t \rangle \langle X_s, X_t \rangle$$

$$+ \sum_{\ell \in [p] \backslash s \cup t} \langle X_s, X_\ell \rangle^3 \langle X_\ell, X_t \rangle + \sum_{\ell \in [p] \backslash s \cup t} \langle X_t, X_\ell \rangle^3 \langle X_\ell, X_s \rangle$$

By equation (P3), we have that

$$\sum_{\ell \in [p] \backslash s \cup t} \langle X_s, X_\ell \rangle^3 \langle X_\ell, X_t \rangle \leq \widetilde{O}(p n^{1.5})$$

Since $\langle X_s, X_t \rangle \leq \widetilde{O}(\sqrt{n})$ and $\sum_{i \in [p]} \langle X_i, X_s \rangle^2 = n^2 + \sum_{i \neq s} \langle X_i, X_s \rangle^2 \leq \widetilde{O}(np)$, we have that

$$\sum_i \langle X_i, X_s \rangle^2 \langle X_s, X_s \rangle \langle X_t, X_s \rangle \leq \widetilde{O}(p n^{2.5})$$

Invoking Lemma 7.11 with $u = X_s$ and $v = X_t$ fixed and view $X_\ell, \ell \in [p] \backslash s \cup t$ as random variables $Z_i$'s, we have that with high probability,

$$\sum_{i \in [p] \backslash s \cup t, \ell \in [p] \backslash s \cup t} \langle X_i, X_\ell \rangle \langle X_i, X_\ell \rangle \langle X_s, X_\ell \rangle \langle X_t, X_\ell \rangle \leq \widetilde{O}(p^2 n^{1.5})$$

Hence combining the three equations above, taking union bound over all choices of $s, t$, we obtain the desired result. $\qquad \square$

**Lemma 7.11.** For $p \geq n$ and a sequence of *good* independent random variables $Z_1, \ldots, Z_p$, and any two fixed vectors $u, v$ with $|u|_\infty \leq \widetilde{O}(1)$ and $|v|_\infty \leq \widetilde{O}(1)$, and $\langle u, v \rangle \leq \widetilde{O}(\sqrt{n})$, we have that with high probability,

$$\sum_{i \in [p]} \sum_{j \in [p]} \langle Z_i, Z_j \rangle^2 \langle Z_j, u \rangle \langle Z_j, v \rangle \leq \widetilde{O}(p^2 n^{1.5})$$

*Proof.* We first extract the consider those cases with $i = j$ separately by expanding

$$\begin{aligned}
\sum_{i \in [p]} \sum_{j \in [p]} \langle Z_i, Z_j \rangle^2 \langle Z_j, u \rangle \langle Z_j, v \rangle &= \sum_{i \neq j} \langle Z_i, Z_j \rangle^2 \langle Z_j, u \rangle \langle Z_j, v \rangle + \sum_i \langle Z_i, Z_i \rangle^2 \langle Z_i, u \rangle \langle Z_i, v \rangle \\
&= \sum_{i \neq j} \langle Z_i, Z_j \rangle^2 \langle Z_j, u \rangle \langle Z_j, v \rangle + \widetilde{O}(pn^{2.5})
\end{aligned} \tag{7.1}$$

where the the last line uses Lemma 7.9. Let $Y_1, \ldots, Y_p$ are independent random variables that have the same distribution as $Z_1, \ldots, Z_p$, respectively, then by Theorem 8.7, we can decouple the sum of functions of $Z_i, jZ_j$ into a sum that of functions of $Z_i$ and $Y_j$,

$$\Pr\left[ \sum_{i \neq j} \langle Z_i, Z_j \rangle^2 \langle Z_j, u \rangle \langle Z_j, v \rangle \geq t \right] \leq C \Pr\left[ \sum_{i \neq j} \langle Y_i, Z_j \rangle^2 \langle Z_j, u \rangle \langle Z_j, v \rangle \geq t/C \right]$$

Now we can invoke Lemma 7.12 which deals with RHS of the equation above, and obtain that with high probability

$$\sum_{i \neq j} \langle Y_i, Z_j \rangle^2 \langle Z_j, u \rangle \langle Z_j, v \rangle \leq \widetilde{O}(p^2 n^{1.5})$$

Therefore, with high probability,

$$\sum_{i \neq j} \langle Z_i, Z_j \rangle^2 \langle Z_j, u \rangle \langle Z_j, v \rangle \leq \widetilde{O}(p^2 n^{1.5})$$

Then combine with equation (7.1) we obtain the desired result. $\qquad\square$

**Lemma 7.12.** For $p \geq n$ and a sequence of *good* independent random variables $Z_1, \ldots, Z_p$, let $Y_1, \ldots, Y_p$ be independent random variables which have the same distribution as $Z_1, \ldots, Z_p$, respectively, then for any two fixed vectors $u, v$ with $|u|_\infty \leq \widetilde{O}(1)$ and $|v|_\infty \leq \widetilde{O}(1)$, and $\langle u, v \rangle \leq \widetilde{O}(\sqrt{n})$, with high probability,

$$\sum_{i \in [p]} \sum_{j \in [p]} \langle Y_i, Z_j \rangle^2 \langle Z_j, u \rangle \langle Z_j, v \rangle \leq \widetilde{O}(p^2 n^{1.5})$$

*Proof.* Let $B = \sum_{i \in [p]} Y_i Y_i^T$. Therefore by Lemma 7.13, we have that with high probability over the randomness of $Y$, $\|B\|_2 \leq \widetilde{O}(p)$, $\text{tr}(B) = pn$. Moreover, by Lemma 7.9, we have that with high probability, $|u^T B v| \leq \widetilde{O}(p\sqrt{n})$. Note that these bounds only depend on the randomness of $Y$, and conditioning on all these bounds are true, we can still use the randomness of $Z_i$'s for concentration. We invoke Lemma 7.14 and obtain that

$$\left| \sum_{i,j \in [p]} \langle Z_j, Y_i \rangle^2 \langle Z_j, u \rangle \langle Z_j, v \rangle \right| = \left| \sum_{j=1}^p Z_j^T B Z_i \langle Z_i, u \rangle \langle Z_i, v \rangle \right| \leq \widetilde{O}(p^2 n^{1.5})$$

$\qquad\square$

**Lemma 7.13.** For $p \geq n$ and a sequence of *good* independent random variables $Z_1, \ldots, Z_p$, we have that with high probability,

$$\left\| \sum_{i \in [p]} Z_i Z_i^T \right\| \leq \widetilde{O}(p)$$

*Proof.* We use matrix Bernstein inequality. First of all, we have that $\mathbb{E}[Z_i Z_i^T] = I_{n \times n}$, and therefore $\mathbb{E}\left[\sum_{i \in [p]} Z_i Z_i^T\right] = p I_{n \times n}$. Moreover, we check the variance of the $Z_i Z_i^T$:

$$\mathbb{E}[Z_i Z_i^T Z_i Z_i^T] = n \, \mathbb{E}[Z_i Z_i^T] = n I_{n \times n}$$

Finally we observe that $\|Z_i Z_i^T\| \leq n$. Thus applying matrix Bernstein inequality we obtain that with high probability,

$$\left\| \sum_{i \in [p]} Z_i Z_i^T - p I_{n \times n} \right\| \leq \widetilde{O}(\sqrt{np} + n) = \widetilde{O}(\sqrt{np})$$

$\square$

**Lemma 7.14.** For $p \geq n$ and a sequence of *good* independent random variables $Z_1, \ldots, Z_p$, and for any fixed symmetric PSD matrix $B \in \mathbb{R}^{n \times n}$ with $\|B\| \leq \widetilde{O}(p)$, $\mathrm{tr}(B) \leq 2pn$, and any two fixed vectors $u, v$ with $|u|_\infty \leq \widetilde{O}(1)$ and $|v|_\infty \leq \widetilde{O}(1)$, and $\langle u, v \rangle \leq \widetilde{O}(\sqrt{n})$, we have that with high probability over the randomness of $Z_i$'s,

$$\left| \sum_{i=1}^{p} Z_i^T B Z_i \langle Z_i, u \rangle \langle Z_i, v \rangle \right| \leq \widetilde{O}(p^2 n^{1.5})$$

*Proof.* Let $W = x^T B x \langle x, u \rangle \langle x, v \rangle$, where $x$ is a random variable that satisfies the conclusion of Proposition 7.2. We first calculate the expectation of $W$,

$$
\begin{aligned}
\mathbb{E}[W] &= \mathbb{E}\left[ \left( \sum_i B_{ii} x_i^2 + \sum_{i \neq j} x_i x_j B_{ij} \right) \left( \sum_i u_i v_i x_i^2 + \sum_{i \neq j} x_i x_j u_i v_j \right) \right] \\
&= (C_4 - C_{2,2}) \sum_i B_{ii} u_i v_i + C_{2,2} \mathrm{tr}(B) \langle u, v \rangle + \mathbb{E}\left[ \sum_{i \neq j} B_{ij} (u_i v_j + u_j v_i) x_i^2 x_j^2 \right] \\
&= (C_4 - 3C_{2,2}) \sum_i B_{ii} u_i v_i + \mathrm{tr}(B) \langle u, v \rangle
\end{aligned}
$$

Therefore by the fact that $|u|_\infty \leq \widetilde{O}(1)$ and $\mathrm{tr}(B) \leq 2pn$, we obtain that $|\mathbb{E}[W]| \leq \widetilde{O}(pn^{1.5})$.

Observe that $Z_j^T B Z_j \leq \widetilde{O}(pn)$ a.s. (with respect to the randomness of $Z_j$), and $\|\langle Z_i, u \rangle \langle Z_i, v \rangle\|_{\psi_1} \leq O(n)$, therefore we have that $\|Z_j^T B Z_j \langle Z_i, u \rangle \langle Z_i, v \rangle\|_{\psi_1} \leq O(pn^2)$. Using Theorem 8.4, we obtain that with high probability,

$$\left| \sum_{i=1}^{p} Z_i^T B Z_i \langle Z_i, u \rangle \langle Z_i, v \rangle - \mathbb{E}\left[ \sum_{i=1}^{p} Z_i^T B Z_i \langle Z_i, u \rangle \langle Z_i, v \rangle \right] \right| \leq \widetilde{O}(n^2 p^{1.5})$$

Using the fact that $\mathbb{E}\left[Z_i^T B Z_i \langle Z_i, u \rangle \langle Z_i, v \rangle\right] \leq \widetilde{O}(pn^{1.5})$ we obtain that with high probability

$$\left|\sum_{i=1}^{p} Z_i^T B Z_i \langle Z_i, u \rangle \langle Z_i, v \rangle\right| \leq \widetilde{O}(p^2 n^{1.5})$$

$\square$

**Lemma 7.15.** Suppose $p \geq n$ and $X_1, \ldots, X_p$ are *good* independent random variables, then with high probability,

$$\|XX^T\|_F^2 \geq (1 - o(1))p^2 n$$

*Proof.* We first $i$ and examine $\sum_{j \neq i} \langle X_j, X_i \rangle^2$ first. We have that $\mathbb{E}[\sum_{j \neq i} \langle X_j, X_i \rangle^2] = (p-1)\|X_i\|^2 = (p-1)n$. Moreover, $\|\langle X_j, X_i \rangle^2\|_{\psi_1} \leq O(n)$ (where $X_j$ is viewed as random and $X_i$ is viewed as fixed). Therefore by Theorem 8.4, we obtain that with high probability over the randomness of $X_j$'s, $(j \neq i)$, $\sum_{j \neq i} \langle X_j, X_i \rangle^2 = (p-1)n \pm \widetilde{O}(n\sqrt{p}) = (1 \pm o(1))pn$. Therefore taking union bound over all $i$, and taking the sum we obtain that

$$\|XX^T\|_F^2 \geq \sum_i \sum_{j \neq i} \langle X_j, X_i \rangle^2 \geq (1 - o(1))p^2 n$$

$\square$

# 8    Toolbox

This section contains a collection of known technical results which are useful in proving the concentration bounds of Section 7. We note that when the data matrix $X$ takes uniformly $\{\pm 1\}$ entries, then $X$ satisfies Proposition 7.2 without any normalization and actually due to the independence of the entries, it's much easier to prove that it satisfies Condition 4.1.

**Definition 8.1** (Orlicz norm $\|\cdot\|_{\psi_\alpha}$). For $1 \leq \alpha < \infty$, let $\psi_\alpha(x) = \exp(x^\alpha) - 1$. For $0 < \alpha < 1$, let $\psi_\alpha(x) = x^\alpha - 1$ for large enough $x \geq x_\alpha$, and $\psi_\alpha$ is linear in $[0, x_\alpha]$. The Orlicz norm $\psi_\alpha$ of a random variable $X$ is defined as

$$\|X\|_{\psi_\alpha} \triangleq \inf\{c \in (0, \infty) \mid \mathbb{E}\left[\psi_\alpha(|X|/c) \leq 1\right] \tag{8.1}$$

Note that by definition $\psi_\alpha$ is convex and increasing. The following Theorem of Ledoux and Talagrand's is our main tool for proving concentration inequalities in Section 7.

**Theorem 8.2** (Theorem 6.21 of [LT13]). There exists a constant $K_\alpha$ depending on $\alpha$ such that for a sequence of independent mean zero random variables $X_1, \ldots, X_n$ in $L_{\psi_\alpha}$, if $0 < \alpha \leq 1$,

$$\left\|\sum_i X_i\right\|_{\psi_\alpha} \leq K_\alpha \left(\left\|\sum_i X_i\right\|_1 + \left\|\max_i \|X_i\|\right\|_{\psi_\alpha}\right) \tag{8.2}$$

and if $1 < \alpha \leq 2$,

$$\left\|\sum_i X_i\right\|_{\psi_\alpha} \leq K_\alpha \left(\left\|\sum_i X_i\right\|_1 + (\sum_i \|X_i\|_{\psi_\alpha}^\beta)^{1/\beta}\right) \tag{8.3}$$

where $1/\alpha + 1/\beta = 1$.

The following convenient Lemma allows us to control the second part of RHS of (8.2) easily.

**Lemma 8.3** ([vdVW00])**.** There exists absolute constant $c$, such that for any real valued random variables $X_1, \ldots, X_n$, we have that

$$\left\| \max_{1 \leq i \leq n} |X_i| \right\|_{\psi_\alpha} \leq c \psi_\alpha^{-1}(n) \max_{1 \leq i \leq n} \|X_i\|_{\psi_\alpha}$$

Using Lemma 8.3 and Theorem 8.2, we obtain straightforwardly the following theorem that will be used many times for proving concentration bounds in this paper.

**Theorem 8.4.** For any $0 < \alpha \leq 1$, there exists a constant $K_\alpha$ such that for a sequence of independent random variables $X_1, \ldots, X_n$,

$$\left\| \sum_i X_i - \mathbb{E}[\sum_i X_i] \right\|_{\psi_\alpha} \leq K_\alpha \sqrt{n} \log n \cdot \max_i \|X_i\|_{\psi_\alpha} \tag{8.4}$$

which implies that with high probability over the randomness of $X_i$'s,

$$\left| \sum_i X_i - \mathbb{E}[\sum_i X_i] \right| \leq \widetilde{O}(K_\alpha \sqrt{n} \cdot \max_i \|X_i\|_{\psi_\alpha})$$

The following two lemmas are used to bound the Orlicz norms of random variables.

**Lemma 8.5.** There exists constant $D_\alpha$ depending on $\alpha$ such that, if two (possibly correlated) random variables $X$, $Y$ have $\psi_\alpha$ Orlicz norm bounded by $\|X\|_{\psi_\alpha} \leq a$ and $\|Y\|_{\psi_\alpha} \leq b$ then $\|XY\|_{\psi_{\alpha/2}} \leq D_\alpha ab$

*Proof.* For any $x, y, a, b, \alpha > 0$,

$$\exp\left(|xy|\right)^{\alpha/2} - 1 \leq \exp\left(\frac{1}{2}|x|^\alpha + \frac{1}{2}|y|^\alpha\right) - 1$$
$$\leq \frac{1}{2}\left((\exp|x|^\alpha - 1) + (\exp|y|^\alpha - 1)\right)$$

Moreover, note that by definition of $\psi_\alpha$, there exists constant $C_\alpha$ and $C_\alpha'$ such that for $x \geq 0$, $C_\alpha'(\exp(x^\alpha) - 1) \geq \psi_\alpha(x) \geq C_\alpha(\exp(x^\alpha) - 1)$. Therefore we have that there exists a constant $E_\alpha$ such that $\psi_{\alpha/2}(|xy|) \leq \frac{E_\alpha}{2}(\psi_\alpha(|x|) + \psi_\alpha(|y|))$. Also note that for any constant $c$, there exsits constant $c'$ such that $\psi_\alpha(x/c') \leq \psi_\alpha(x)/c$. Therefore, choosing $D_\alpha$ such that $\psi_{\alpha/2}(x/D_\alpha) \leq \psi_\alpha(x)/E_\alpha$ for all $x \geq 0$ we obtain that

$$\mathbb{E}\left[\psi_{\alpha/2}(\frac{|XY|}{abD_\alpha})\right] \leq \mathbb{E}\left[\psi_{\alpha/2}(\frac{|XY|}{ab})\right]/E_\alpha \leq \frac{1}{2}\left(\mathbb{E}[\psi_\alpha(|X|/a)] + \mathbb{E}[\psi_\alpha(|Y|/b)]\right) \leq 1$$

$\square$

**Lemma 8.6.** Suppose random variable $X$ has $\psi_\alpha$-Orlicz norm $a$, then $X - \mathbb{E}[X]$ has $\psi_\alpha$ Orlicz norm at most $2a$.

*Proof.* First of all, since $\psi_\alpha$ is convex and increasing on $[0, \infty)$, we have that $\mathbb{E}[\psi_\alpha(|X|/a)] \geq \psi_\alpha(\mathbb{E}[|X|]/a) \geq \psi_\alpha(|\mathbb{E}[X]|/a)$. Then we have that

$$\mathbb{E}\left[\psi_\alpha(\frac{|X - \mathbb{E}[X]|}{2a})\right] \leq \mathbb{E}[\psi_\alpha(\frac{|X|}{2a} + \frac{|\mathbb{E}[X]|}{2a})] \leq \mathbb{E}\left[\frac{1}{2}\psi_\alpha(|X|/a) + \frac{1}{2}\psi_\alpha(|\mathbb{E}[X]|/a)\right] \leq \mathbb{E}[\psi_\alpha(|X|/a)] \leq 1$$

where we used the convexity of $\psi_\alpha$ and the fact that $\mathbb{E}[\psi_\alpha(|X|/a)] \geq \psi_\alpha(|\mathbb{E}[X]|/a)$ $\qquad\square$

The following Theorem of [PMS95] is useful to decouple the randomness of a sum of correlated random variables into a form that is easier to control.

**Theorem 8.7** (Special case of Theorem 1 of [PMS95])**.** Let $X_1, \ldots, X_n, Y_1, \ldots, Y_n$ are independent random variables on a measurable space over $S$, where $X_i$ and $Y_i$ has the same distribution for $i = 1, \ldots, n$. Let $f_{ij}(\cdot, \cdot)$ be a family of functions taking $S \times S$ to a Banach space $(B, \|\cdot\|)$. Then there exists absolute constant $C$, such that for all $n \geq 2$, $t > 0$,

$$\Pr\left[\left\|\sum_{i \neq j} f_{ij}(X_i, X_j)\right\| \geq t\right] \leq C \Pr\left[\left\|\sum_{i \neq j} f_{ij}(X_i, Y_j)\right\| \geq t/C\right]$$

The following lemma provides a simple way to prove the PSDness of a matrix that has large value on the diagonal and small off-diagonal values.

**Lemma 8.8** (Consequence of Gershgorin Circle Theorem)**.** Suppose a matrix $\Gamma$ is of the form $\Gamma = \begin{bmatrix} A & B \\ C & D \end{bmatrix}$ where $A, D$ are square diagonal matrices, and $C$ is of dimension $n \times m$. Then $\Gamma$ is PSD if there exists $\alpha > 0$ such that the following holds: $A_{ii} \geq \frac{1}{\alpha}\sum_{j \in [n]}|C_{ij}|, \forall \in [p]$ and $D_{jj} \geq \alpha\sum_{i \in [m]}|C_{ij}|, \forall j \in [p]$.

*Proof.* Let vector $u = (\alpha\mathbf{1}_m, \alpha^{-1}\mathbf{1}_n)$ and $v = (\alpha^{-1}\mathbf{1}_m, \alpha\mathbf{1}_n)$, where $\mathbf{1}_n$ is $n$-dimensional all 1's vector. Then $\Gamma$ can be written as $\Gamma = vv^T \odot (uu^T \odot \Gamma)$, where $\odot$ denotes the entries-wise product of two matrices (That is, $A \odot B$ is a matrix with entry $A_{ij}B_{ij}$). Using the Gershgorin Circle Theorem and the conditions of the Lemma we obtain that $uu^T \odot \Gamma$ is PSD and therefore $\Gamma$ is PSD. $\qquad\square$

# 9 Conclusions and future directions

In this paper we prove a lower bounds on the number of samples required to solve the Sparse PCA problem by degree-4 SoS algorithms. This extends the (spectral) degree-2 SoS lower bound for the problem, establishing the quadratic gap from the number of samples required by the (inefficient) information theoretic bound. It remains an interesting problem to extend our lower bounds to higher degree SoS algorithms (or even better, show that with some constant degree, one can solve the problem with fewer samples). One specific difficulty we encountered in trying to extend the lower bound to higher degree was the polynomial constraint $x_i^3 = x_i$, capturing the discreteness of the hidden sparse vector. The SoS formulation of the problem without this condition is interesting as well, and lower bound for it may be easier.

As mentioned, it is possible that the best way to prove strong SoS lower bounds for Sparse PCA is via the reduction of Berthet and Rigollet's [BR13a], namely by improving existing lower bounds for the Planted Clique problem. However, we note that this approach is limited as well, as it seems

that sparse PCA is significantly harder. Specifically, Planted Clique has a simple $O(\log n)$-degree SoS algorithm (and thus a quasi-polynomial time) *optimal* solution, whereas for Sparse PCA we know of no better sample-optimal algorithm than one running in exponential $p^{O(k)}$ time. It is thus conceivable that one can even prove $\Omega(k)$-degree SoS lower bounds for this problem.

More generally, we believe that statistical and machine learning problems provide a new and challenging setting for testing the power and limits and SoS algorithms. While we have fairly strong techniques for proving optimal SoS lower bounds for combinatorial optimization problems, we lack similar ones for ML problems. In particular, many other problems besides Sparse PCA seem to exhibit the apparent trade-off between the number of samples required information theoretically versus via computationally efficient techniques, offering fertile ground for attempting SoS lower bounds establishing such trade-offs.

Finally it would be nice to see more reductions between problems of statistical and ML nature, as the one by [BR13a]. Efficient reductions have proved extremely powerful in computational complexity theory and optimization, enabling the framework of complexity classes and complete problems. Creating such a framework within machine learning will hopefully expose structure on the relative difficulty of problems in this vast area, highlighting some problems as more central to attack, and enabling both new algorithms and new lower bounds.

**Acknowledgments:** We would like to thank Sanjeev Arora, Boaz Barak, Philippe Rigollet and David Steurer for helpful discussions throughout various stages of this work.

## Footnotes

[1]We treat $\lambda$ as a constant so that we omit the dependence on it for simplicity throughout the introduction section

[2]An average case version of the Clique problem in which the input is a random graph in which a much larger than expected clique is planted.

[3] Or we assume that they go to infinity as typically done in statistics.

[4] A real random variable $X$ is subgaussian with variance proxy $\sigma^2$ if it has similar tail behavior as gaussian distribution with variance $\sigma^2$. More formally, if for any $t \in \mathbb{R}$, $\mathbb{E}[\exp(tX)] \leq \exp(t^2\sigma^2/2)$

[5] $\Lambda'$ is index by $ii$, $i = 1, \ldots, p$

[6] A real random variable $X$ is subgaussian with variance proxy $\sigma^2$ if it has similar tail behavior as gaussian distribution with variance $\sigma^2$, and formally if for any $t \in \mathbb{R}$, $\mathbb{E}[\exp(tX)] \leq \exp(t^2\sigma^2/2)$