[Reviews · NeurIPS 2015]

Submitted by Assigned_Reviewer_1

The paper applies the recently trending idea of SoS relaxations to the problem of sparse PCA, which is a mainstream ML topic. The basic idea of SoS relaxations is to convert the optimization problem into a problem over distributions (rather than individual solutions) and to constrain the distributions using "moment-like" constraints. By using only second-order moments, one obtains an SoS2 relaxation, and fourth-order moments give SoS4. The main result of the paper is to show that SoS4 is not much tighter than SoS2 for sparse PCA.

The paper is well written and may help to introduce the SoS techniques to the ML community. On the other hand, the analysis follows closely the analysis for the planted clique problem (which is not surprising given that planted clique can also be written as a discrete Rayleigh quotient problem).
Summary: Well written paper that may introduce the SoS methods to the NIPS community.

Submitted by Assigned_Reviewer_2

The paper has a useful overview of the SOS framework, sparse PCA and Lasserre relaxations. The results are quite interesting and novel, however, they might be more of a focus

for the CS theory community rather than the NIPS audience. In particular, the impossibility result for degree 4 does not say anything about higher degrees -- and the Lasserre hierarchy may still be practical for degree 6 and maybe higher.

The paper would benefit from a proof-reading, there are multiple typos and/or grammar mistakes.

Just one example: line 111: There has been an impressive __?

of SoS degree lower bounds
Summary: The paper considers SOS relaxations for the sparse PCA problem. In previous work it has been established that 2-nd order SOS can not close the gap between information-theoretic lower bounds and efficient algorithms in terms of sample complexity. This paper extends this result to 4-th degree SOS polynomials, but doesnt't address higher-order

degrees.

Submitted by Assigned_Reviewer_3

Paper 997 - Sum-of-Squares Lower Bounds for Sparse PCA ======================================================

The authors explore the statistical-computational trade-off in the Sparse PCA

problem within the Sum-Of-Squares convex relaxation hierarchy.

Their work

nicely complements many recent papers in the literature on this subject and is

most directly related to the paper by Krauthgamer, Nadler, and Vilenchik (2015). The results of KNV (2015) can be interpreted as a negative result for the SoS

degree-2 relaxation.

This paper provides analogous results for the degree-4 SoS relaxation: showing that the optimal value of the the program is large with high probability, and hence prone to false rejection, and that its output is

weakly correlated with the true.

This paper is closer to the COLT community,

but the results are very good and the paper is well-written.

It should be

accepted into NIPS.

I only have a few typographical corrections:

page 3, line 140: "even to" -> "even" page 3, line 145: "An very" -> "A very" page 5, line 234: "once can" -> "one can" page 6, line 299: "above as ." -> "above." page 6, line 310: "Rigollet's" -> "Berthet and Rigollet's"
Summary: This paper is closer to the COLT community, but the results are very good and the paper is well-written.

It should be

accepted into NIPS.

Author Feedback
Author rebuttal: We thank the reviewers for the detailed reviews. We highlight the contributions of the paper below, which will clarify the questions and concerns of the reviewers.

1. Our deg-4 SoS lower bound does not follow from combining the Berthet and Rigollet(2013) reduction from planted clique to sparse PCA, and the best known planted clique lower bounds available, Deshpande and Montanari(2015). In fact the later two together imply only a trivial lower bound (namely, the minimax lower bound) for SoS algorithms. (A little bit more detail below)

2. A week ago new planted clique lower bounds were announced, from which the [BR] will imply a degree-4 SoS lower bound - however, as the [BR] reduction changes the distribution, it would not, e.g., apply to Gaussian signal or noise, as our result does.

3. To address a concern of reviewer 7. The lower bound technique, namely the moments used in our lower bound proof, are actually very different than the ones in the planted clique lower bounds of Meka, Potechin, Wigderson(2015) and Deshpande and Montanari(2015). The analysis as well is very different. Of course, in all these proofs one has to ``guess'' moments, and then prove somehow that the resulting moment matrix is PSD with high probability over the input distribution. But there are no general tools, and the few existing lower bounds so far are quite specific to the problems. Indeed, one hope from finding more such lower bounds is the emergence of general limitations on the powerful SoS technique for solving such statistical problems.

4. To address a concern of reviewer 5: Our model assumptions are simplistic, but we are proving impossibility results, and simplistic assumptions make the result stronger. Because we picked the simplest, most restrictive version of sparse PCA, our lower bounds apply to other, more realistic models (like general covariance models, multi-spike models, etc), showing that it is also impossible to achieve better sample complexity for them using a SoS-4 algorithms.

5. To address a concern of reviewer 1, on suitability of the paper to the TCS vs. ML communities. We feel the paper is appropriate for both. We submitted to NIPS due to the special importance of the sparse PCA problem to the ML community, and the fact that using SoS algorithms were shown successful recently in addressing other problems in which sparsity is a concern. We show this is not the case here, at least for degree-4 (namely 4 rounds) of SoS. Needless to say, extending our results to higher degrees is a natural and major challenge. Indeed, one way to see the difference of our proof from the planted clique lower bounds is that there extending the moments and proofs to higher degrees is natural, whereas we saw no such extension here.
To expand on our Remark 1 above. We proved unconditionally that using deg-4 SoS, the sample complexity of sparse PCA is at least n >= k^2. The reduction of BR basically says that an n^b lower bound for planted clique is (n is the size of the graph here), implies the sample complexity for sparse PCA will be at least n >= k^{b/(4b-1)} (where n is the number of samples and k is the sparsity). If one plugs in the well-believed (and announced last week) value b = 1/2, then the resulting bound for sparse PCA is n >= k^2 as desired (albeit only under some natural distributions on signal and noise, not in the generality we get). However, the best available lower bound for planted clique for deg-4 SoS by Deshpande and Montanari(2015) gives b = 1/3, and this implies the sample complexity for sparse PCA is at least n >= k, which is merely just the minimax lower bound without any computational constraints.

Finally, we will revise the typos and format issues, according to the reviewers' suggestions.